# MoCo-EA: Exploiting Adversarial Mode Connectivity for Efficient Evolutionary Attacks

Hyo Seo Kim [1]  Gang Luo [1]  Can Chen [2]  Binghui Wang [1]  Yue Duan [3]  Ren Wang [1]

## Abstract

Evolutionary algorithms for adversarial attacks leverage population-based search to discover perturbations without gradient information, but suffer from inefficient crossover operations that destroy adversarial properties through discrete interpolation. We introduce Mode Connectivity Evolutionary Attack (MoCo-EA), which replaces traditional crossover with a novel Bézier crossover operator that optimizes perturbations along a continuous Bézier curve between parent perturbations. Our key insight is that adversarial examples lie on connected manifolds where intermediate points maintain and often enhance attack effectiveness. We demonstrate three findings: (1) Successful adversarial perturbations exhibit mode connectivity; (2) Intermediate points along optimized paths achieve higher transferability than endpoints; (3) Bézier crossover dramatically outperforms discrete genetic operations while reducing convergence time and query requirements. By exploiting the geometric structure of adversarial space through path optimization, MoCo-EA provides an efficient and reliable method. Our work challenges the traditional view of adversarial examples as isolated points and opens new directions for both attack generation and defense research.

## 1. Introduction

Adversarial attacks expose the vulnerability of deep neural networks to carefully crafted input perturbations (Goodfellow et al., 2015; Madry et al., 2018; Wang et al., 2022b; Li et al., 2023). These attacks are broadly categorized as white-box or black-box, depending on the attacker's access to the model (Costa et al., 2024). White-box attacks leverage full access to model parameters and gradients (Goodfellow et al., 2015; Carlini & Wagner, 2017; Madry et al., 2018; Croce & Hein, 2020). These optimize perturbations under $\ell_p$-norm constraints and remain state-of-the-art in bounded settings. In contrast, black-box attacks rely on model queries or surrogates, including score-based (Chen et al., 2017), decision-based (Brendel et al., 2018; Chen et al., 2020), and evolutionary methods (Alzantot et al., 2019; Su et al., 2019). Among these, evolutionary algorithms constitute a distinct category that operates without gradient information, offering inherent parallelizability for effective exploration and population diversity to escape local optima. They typically evolve a population of perturbations using operators like mutation, selection, and crossover, mimicking biological evolution.

However, evolutionary attacks are rarely explored in the white-box setting, despite the fact that white-box access can provide valuable signals to improve population-based search. Existing methods such as GenAttack are purely gradient-free and rely on element-wise crossover operations that ignore the underlying geometry of the input space (Alzantot et al., 2019), resulting in inefficiencies, poor transferability, and limited diversity. This motivates the need to revisit evolutionary attacks in the white-box setting. While several state-of-the-art gradient-based attacks already exist, they typically follow a single optimization trajectory, which limits exploration and makes them vulnerable when gradients are unreliable or obfuscated.

To address these limitations, we identify and exploit adversarial mode connectivity, a previously unexplored property defined as the existence of continuous paths between different adversarial perturbations that maintain attack effectiveness throughout. While mode connectivity has been extensively studied in neural network parameter spaces (Freeman & Bruna, 2017; Draxler et al., 2018; Garipov et al., 2018; Wang et al., 2023; Ren et al., 2025; Shi & Wang, 2026), its application to bridging adversarial input perturbations remains unexplored. We demonstrate that successful adversarial examples lie on connected manifolds where continuous paths preserve adversarial properties. More importantly, we find that intermediate points along optimized paths exhibit significantly higher transferability than endpoints, with sub-

[1]Illinois Institute of Technology [2]University of North Carolina at Chapel Hill [3]Singapore Management University. Correspondence to: Ren Wang <rwang74@illinoistech.edu>.

*Proceedings of the 43rd International Conference on Machine Learning*, Seoul, South Korea. PMLR 306, 2026. Copyright 2026 by the author(s).

stantial improvements in attack success and rescue rates for previously failed attacks. This discovery reveals that the adversarial space has rich geometric structure amenable to continuous exploration rather than discrete sampling. Related advances in input-space connectivity further highlight this potential (Vrabel et al., 2025).

Building on these insights, we propose Mode Connectivity Evolutionary Attack (MoCo-EA), which fundamentally reimagines crossover through continuous path optimization. Our approach systematically studies adversarial perturbation connectivity, showing that successful attacks are not isolated points but lie on connected adversarial manifolds. We further reveal that intermediate points on optimized Bézier curves achieve stronger and more transferable attacks than endpoints. Finally, we develop MoCo-EA, replacing discrete crossover with Bézier curve interpolation, achieving near-perfect success while sharply reducing convergence time and query requirements. An overview of the proposed MoCo-EA is shown in Figure 1. In brief, from two parent perturbations, we optimize a quadratic Bézier path in perturbation space to connect those parents, evaluate the resulting connectivity under progressively harder settings and with multi-image augmentation, and finally instantiate a geometry-aware evolutionary attack that employs a Bézier crossover operator.

We summarize our contributions below:

- We provide the first systematic study of adversarial perturbation connectivity, showing that successful attacks are not isolated points but are connected by continuous paths that preserve adversarial effectiveness.

- We reveal that intermediate points on optimized Bézier paths between parents are stronger than endpoints, with transferability that generally increases as more auxiliary images guide path optimization.

- We develop MoCo-EA, a geometry-aware evolutionary attack that replaces discrete crossover with Bézier crossover, achieving near-perfect success across norms while sharply cutting convergence time and query complexity.

## 2. Related Work

**Gradient-based adversarial attacks.** Gradient-based attacks are the most widely used methods in the white-box setting, where the attacker has full access to model parameters and gradients (Zhang et al., 2025; Wang et al., 2021). The single-step Fast Gradient Sign Method (FGSM) introduced by Goodfellow et al. (2015) and its multi-step variant, PGD (Madry et al., 2018), became standard white-box attacks and the backbone of adversarial training. Optimization-based

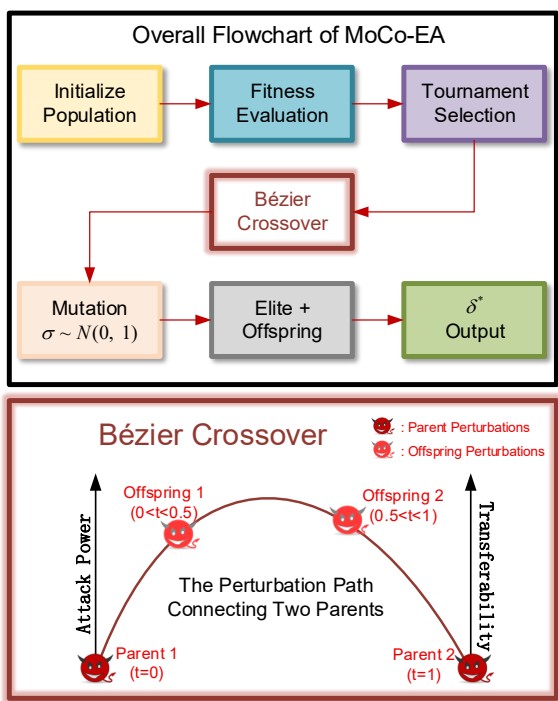

*Figure 1.* Overview of MoCo-EA.

C&W attacks target different $\ell_p$ norms (Carlini & Wagner, 2017), while AutoAttack provides a robust ensemble for evaluation (Croce & Hein, 2020). Stronger transfer-based variants incorporate momentum (Dong et al., 2018), input diversity (Xie et al., 2019), and translation invariance (Dong et al., 2019). However, most gradient-based methods operate locally around a single example and do not explicitly leverage global structure that might connect different adversarial modes, motivating our study of continuous connectivity between successful perturbations.

**Evolutionary adversarial attacks.** Evolutionary algorithms (Wang et al., 2022a) have proven effective in black-box scenarios due to their ability to explore complex and non-differentiable search spaces. Representative methods include GenAttack (Alzantot et al., 2019) and the one-pixel differential-evolution attack (Su et al., 2019). Subsequent work improved sampling/guidance via probability-guided genetic search (Chen et al., 2019) and gradient/score estimation. However, they remain underexplored in white-box settings, where gradient information could enhance evolutionary dynamics. Most existing evolutionary attacks use element-wise crossover and heuristic mutations, which are agnostic to the structure of the data manifold or loss surface. Moreover, simply incorporating gradients into mutation steps or fitness scoring may lead to instability or

mode collapse. Our work departs from these approaches by proposing a structured and geometry-aware evolutionary attack that explicitly models the connectivity between adversarial examples. Unlike prior methods that treat perturbations as isolated points in input space, we introduce Bézier-based crossover that leverages gradient-informed path optimization to interpolate through adversarial modes. This enables us to preserve adversarial properties along the path while improving sample diversity and transferability.

## 3. Adversarial Mode Connectivity

### 3.1. Bézier Curve for Mode Connectivity

Our approach begins by identifying two distinct adversarial perturbations that serve as endpoints for our Bézier curve construction. We employ PGD (Madry et al., 2018), which remains one of the strongest first-order adversarial attack methods. Given a clean image $\mathbf{x} \in [0, 1]^d$ with true label $\mathbf{y}$, and a classifier $f_{\boldsymbol{\theta}}$, PGD optimizes:

$$\max_{\|\boldsymbol{\delta}\|_p \leq \epsilon} \mathcal{L}(f_{\boldsymbol{\theta}}(\mathbf{x} + \boldsymbol{\delta}), \mathbf{y}), \qquad (1)$$

where $\mathcal{L}$ is the cross-entropy loss, and $\| \cdot \|_p$ constrains the perturbation within an $\ell_p$-ball of radius $\epsilon$. To obtain two distinct local optima $\boldsymbol{\delta}_1$ and $\boldsymbol{\delta}_2$, we run PGD twice with different random initializations, each starting from a random perturbation sampled within the $\ell_p$ ball using different seeds. Both $\boldsymbol{\delta}_1$ and $\boldsymbol{\delta}_2$ fool the classifier (i.e., $f_{\boldsymbol{\theta}}(\mathbf{x} + \boldsymbol{\delta}_1) \neq \mathbf{y}$ and $f_{\boldsymbol{\theta}}(\mathbf{x} + \boldsymbol{\delta}_2) \neq \mathbf{y}$) while capturing different adversarial patterns in the perturbation space.

Adversarial solutions obtained from different initializations often correspond to different local optima, yet they can be connected in the adversarial perturbation space by a path along which predictions remain adversarial. We use mode connectivity to explicitly search such a path because it exposes shared structure among adversarial solutions and helps maintain adversarial effectiveness along the whole trajectory. Among possible parameterizations, we adopt a quadratic Bézier curve, which is widely used in mode-connectivity settings and provides an efficient and effective two-endpoint parameterization with a single learnable control point. Quadratic Bézier curves also offer a favorable expressivity–stability trade-off. They are flexible enough to capture curved high-loss adversarial connections while remaining amenable to stable optimization and norm-ball projection. In contrast to discrete element-wise crossover, which often disrupts adversarial structure, Bézier crossover preserves coherence by construction and can be optimized with only a few gradient steps.

Given two adversarial endpoints $\boldsymbol{\delta}_1$ and $\boldsymbol{\delta}_2$, we define the quadratic Bézier curve as follows:

$$\mathbf{B}(t; \boldsymbol{\delta}_c) = (1-t)^2 \boldsymbol{\delta}_1 + 2(1-t)t\boldsymbol{\delta}_c + t^2 \boldsymbol{\delta}_2, \ t \in [0, 1] \quad (2)$$

where $\boldsymbol{\delta}_c$ is the learnable control point that determines the curvature of the path.

We initialize $\boldsymbol{\delta}_c^{(0)} = \frac{1}{2}(\boldsymbol{\delta}_1 + \boldsymbol{\delta}_2)$ and optimize it by maximizing adversarial loss along the curve:

$$\boldsymbol{\delta}_c^* = \arg\max_{\boldsymbol{\delta}_c} \ \mathbb{E}_t \Big[ \mathcal{L}\big(f_{\boldsymbol{\theta}}\big(\mathbf{x} + \Pi_\epsilon \left[ \mathbf{B}(t; \boldsymbol{\delta}_c) \right]\big), \mathbf{y}\big) \Big], \quad (3)$$

where $\Pi_\epsilon$ denotes projection onto the $\ell_p$-ball of radius $\epsilon$, and the simplified notation $\mathbb{E}_t$ denotes $\mathbb{E}_{t \sim \mathcal{U}(0,1)}$.

This framework allows us to optimize entire perturbation paths rather than isolated endpoints, enabling a direct examination of the structure of adversarial perturbations in subsequent sections. Classical mode-connectivity studies show that distinct solutions can often be linked by smooth paths that preserve low loss (Garipov et al., 2018; Wang et al., 2024). In our setting, we extend this structural idea to the space of adversarial perturbations, but with the objective inverted. Rather than maintaining low loss, we search for continuous trajectories along which the adversarial loss remains high. Appendix A provides a geometric intuition for ReLU networks.

### 3.2. Connectivity Settings

PGD optimizes a single perturbation for one image and therefore tends to converge to sharp, highly localized adversarial maxima with limited transferability (Qin et al., 2022). In contrast, optimizing the Bézier control point requires maintaining high loss at multiple sampled points along the curve, which prevents the solution from collapsing onto such sharp maxima. This multi-point objective encourages the entire trajectory to move toward flatter and more stable high-loss regions, where perturbations generally exhibit stronger cross-instance generalization.

We adopt three settings to systematically study adversarial mode connectivity under different levels of generalization. Setting A focuses on a single image, ensuring the feasibility of connecting two adversarial modes in the simplest case. Setting B extends to multiple images of the same class, testing whether a single curve parameter $\boldsymbol{\delta}_c$ can generalize across variations in appearance while keeping the label fixed. Setting C considers images from different classes, which is the most challenging scenario due to greater semantic dissimilarity.

**Setting A (Image-wise Connectivity).** For a single image $\mathbf{x}$ with label $\mathbf{y}$, we find $\boldsymbol{\delta}_1, \boldsymbol{\delta}_2$ via PGD on the same image and optimize

$$\boldsymbol{\delta}_c^* = \arg\max_{\boldsymbol{\delta}_c} \ \mathbb{E}_t \Big[ \mathcal{L}\big(f_{\boldsymbol{\theta}}(\mathbf{x} + \Pi_\epsilon [ \mathbf{B}(t; \boldsymbol{\delta}_c) ]), \mathbf{y}\big) \Big]. \quad (4)$$

This setting evaluates whether adversarial modes for a single image can be connected while still maintaining attack success.

**Setting B (Class-wise Connectivity).** Given two images $\mathbf{x}_1, \mathbf{x}_2$ from the same class $\mathbf{y}$, we compute $\boldsymbol{\delta}_1 = \mathrm{PGD}(\mathbf{x}_1, \mathbf{y})$ and $\boldsymbol{\delta}_2 = \mathrm{PGD}(\mathbf{x}_2, \mathbf{y})$, then optimize

$$\boldsymbol{\delta}_c^* = \arg \max_{\boldsymbol{\delta}_c} \frac{1}{2} \sum_{i=1}^{2} \mathbb{E}_t \Big[ \mathcal{L}\big(f_{\boldsymbol{\theta}}(\mathbf{x}_i + \Pi_\epsilon[\mathbf{B}(t; \boldsymbol{\delta}_c)]), \mathbf{y}\big) \Big]. \quad (5)$$

This setting evaluates whether adversarial perturbations discovered on different samples of the same class can be connected in a unified curve.

**Setting C (Cross-class Connectivity).** For images $\mathbf{x}_1, \mathbf{x}_2$ from different classes $\mathbf{y}_1, \mathbf{y}_2$, with $\boldsymbol{\delta}_1 = \mathrm{PGD}(\mathbf{x}_1, \mathbf{y}_1)$ and $\boldsymbol{\delta}_2 = \mathrm{PGD}(\mathbf{x}_2, \mathbf{y}_2)$, we optimize

$$\boldsymbol{\delta}_c^* = \arg \max_{\boldsymbol{\delta}_c} \frac{1}{2} \sum_{i=1}^{2} \mathbb{E}_t \Big[ \mathcal{L}\big(f_{\boldsymbol{\theta}}(\mathbf{x}_i + \Pi_\epsilon[\mathbf{B}(t; \boldsymbol{\delta}_c)]), \mathbf{y}_i\big) \Big]. \quad (6)$$

This setting examines whether adversarial connectivity can hold across different semantic classes, which is the most general and challenging scenario.

### 3.3. Multi-image Augmentation for Transferability

To improve the transferability of discovered adversarial curves, we use two kinds of images during optimization. The main images are those used to define the endpoints of the curve. Auxiliary images are additional samples that regularize and improve the transferability of the learned curve. These auxiliary images encourage the curve to encode perturbations that generalize across visual variations. For Setting A we select auxiliary images from the same class as $\mathbf{y}$; for Setting B from the same class $\mathbf{y}$; for Setting C we select a balanced set from the two classes $\mathbf{y}_1$ and $\mathbf{y}_2$. In all cases, auxiliary images are drawn from a held-out pool distinct from training and test splits.

For any image $(\mathbf{x}_k, \mathbf{y}_k)$ (main or auxiliary), the per-image adversarial loss along the curve follows the same construction as in (3):

$$\mathcal{L}_k(t; \boldsymbol{\delta}_c) := \mathcal{L}\big(f_{\boldsymbol{\theta}}\big(\mathbf{x}_k + \Pi_\epsilon[\mathbf{B}(t; \boldsymbol{\delta}_c)]\big), \mathbf{y}_k\big), \quad (7)$$

where $t \sim \mathcal{U}(0, 1)$ and $\mathbf{B}(t; \boldsymbol{\delta}_c)$ is the quadratic Bézier path shared across all images.

Given nonnegative weights $w_{\mathrm{main}}$ and $w_{\mathrm{aux}}$ ($w_{\mathrm{main}} > w_{\mathrm{aux}}$ to emphasize the main images), we optimize

$$
\begin{aligned}
\boldsymbol{\delta}_c^* = \arg \max_{\boldsymbol{\delta}_c} \ \mathbb{E}_t \Big[ &\sum_{i \in \mathrm{main}} w_{\mathrm{main}} \, \mathcal{L}_i^{\mathrm{main}}(t; \boldsymbol{\delta}_c) \\
&+ \sum_{j \in \mathrm{aux}} w_{\mathrm{aux}} \, \mathcal{L}_j^{\mathrm{aux}}(t; \boldsymbol{\delta}_c) \Big],
\end{aligned} \quad (8)
$$

which maximizes adversarial loss along the curve for both main and auxiliary images while respecting the $\ell_p$-budget via the projection operator.

---

**Algorithm 1** Bézier Crossover

**Input:** parents $\boldsymbol{\delta}_1$ and $\boldsymbol{\delta}_2$, image $\mathbf{x}$, label $y$, model $f_{\boldsymbol{\theta}}$
**Parameter:** control-step count $\tau$, step size $\alpha$, projection $\Pi_{\|\cdot\|_p \leq \epsilon}$
**Output:** two offspring perturbations
$\boldsymbol{\delta}_c \leftarrow (\boldsymbol{\delta}_1 + \boldsymbol{\delta}_2)/2$
**for** $step = 1$ **to** $\tau$ **do**
    $loss \leftarrow 0$
    **for** $t \in \{0.25, 0.5, 0.75\}$ **do**
        $\boldsymbol{\delta}_t \leftarrow \mathbf{B}(t; \boldsymbol{\delta}_c)$
        $loss \leftarrow loss - \mathcal{L}\big(f_{\boldsymbol{\theta}}\big(\mathbf{x} + \Pi_{\|\cdot\|_p \leq \epsilon}[\boldsymbol{\delta}_t]\big), y\big)$
    **end for**
    $\boldsymbol{\delta}_c \leftarrow \boldsymbol{\delta}_c - \alpha \cdot \nabla_{\boldsymbol{\delta}_c} loss$
**end for**
$c_1 \leftarrow \mathrm{SelectBest}\big(\{\mathbf{B}(t; \boldsymbol{\delta}_c) \mid t \in (0, 0.5)\}\big)$
$c_2 \leftarrow \mathrm{SelectBest}\big(\{\mathbf{B}(t; \boldsymbol{\delta}_c) \mid t \in (0.5, 1)\}\big)$
**return** $\Pi_{\|\cdot\|_p \leq \epsilon}[c_1]$, $\Pi_{\|\cdot\|_p \leq \epsilon}[c_2]$

---

When auxiliary images are incorporated, the control point $\boldsymbol{\delta}_c$ must further induce high loss across multiple inputs at once. This multi-image objective acts as an implicit regularizer that biases the optimization toward perturbation directions that remain adversarial under larger input variation. As a result, the learned path increasingly aligns with more universal high-loss directions, providing a natural explanation for the strong transferability.

## 4. Mode Connectivity Evolutionary Attack

### 4.1. Traditional Evolutionary Algorithms

We assume a white-box threat model with full knowledge of the model parameters. In this white-box setting we can utilize adversarial mode connectivity to replace the crossover operation used in traditional evolutionary attacks.

Evolutionary algorithms (EAs) have emerged as powerful gradient-free methods for generating adversarial examples, benefiting from population diversity to escape local optima. The traditional EA framework for adversarial attacks operates through iterative population-based optimization. We initialize a population $P^{(0)} = \{\boldsymbol{\delta}_1, \boldsymbol{\delta}_2, \ldots, \boldsymbol{\delta}_N\}$ of $N$ random perturbations within the $\epsilon$-ball, evaluate a fitness score for each perturbation based on attack success, and use tournament selection to choose parent pairs for reproduction. Traditional crossover combines two parent perturbations element-wise, and mutation is implemented by adding Gaussian noise with some probability $p_m$:

$$
\mathrm{child}[j] = \begin{cases} \mathrm{parent}_1[j] & \text{with } \Pr(0.5), \\ \mathrm{parent}_2[j] & \text{otherwise}, \end{cases} \quad \text{(Crossover)} \quad (9)
$$

$$\boldsymbol{\delta}' = \boldsymbol{\delta} + \eta \cdot \mathcal{N}(0, \sigma^2 I) \qquad \text{(Mutation)}.$$

*Table 1.* **Connectivity along Bézier paths.** Results are reported on CIFAR-10 and ImageNet across three settings with 25 data cases each (single images in Setting A and image pairs in Settings B and C). "*ASR1*" and "*ASR2*" denote the fraction of interior points that successfully attack the first and second image, respectively, when defined. "*ASR Both*" denotes the fraction of interior points that successfully attack both images simultaneously. In Setting A, it corresponds to the path success rate for the single image. "*ASR Avg.*" denotes the average of ASR1 and ASR2 when defined, and is set equal to "*ASR Both*" in Setting A. Attacks are evaluated along Bézier curves connecting the endpoints. All values are reported as mean $\pm$ standard deviation.

| Setting | Norm | CIFAR-10 | | | | ImageNet | | | |
|---|---|---|---|---|---|---|---|---|---|
| | | ASR1 | ASR2 | ASR Both | ASR Avg. | ASR1 | ASR2 | ASR Both | ASR Avg. |
| A | $\ell_\infty$ | N/A | N/A | 100.0$\pm$0.0 | 100.0$\pm$0.0 | N/A | N/A | 100.0$\pm$0.0 | 100.0$\pm$0.0 |
| | $\ell_2$ | N/A | N/A | 100.0$\pm$0.0 | 100.0$\pm$0.0 | N/A | N/A | 100.0$\pm$0.0 | 100.0$\pm$0.0 |
| | $\ell_1$ | N/A | N/A | 99.9$\pm$0.4 | 99.9$\pm$0.4 | N/A | N/A | 100.0$\pm$0.0 | 100.0$\pm$0.0 |
| B | $\ell_\infty$ | 98.6$\pm$1.6 | 98.3$\pm$1.8 | 97.0$\pm$2.4 | 98.5$\pm$1.2 | 99.5$\pm$0.9 | 99.4$\pm$0.9 | 98.9$\pm$1.4 | 99.4$\pm$0.7 |
| | $\ell_2$ | 97.8$\pm$1.7 | 97.7$\pm$1.8 | 95.4$\pm$2.9 | 97.7$\pm$1.4 | 99.3$\pm$1.1 | 99.3$\pm$1.1 | 98.6$\pm$1.9 | 99.3$\pm$1.0 |
| | $\ell_1$ | 87.2$\pm$32.0 | 75.0$\pm$42.2 | 62.3$\pm$46.6 | 81.1$\pm$23.4 | 100.0$\pm$0.0 | 99.7$\pm$0.7 | 99.7$\pm$0.7 | 99.8$\pm$0.4 |
| C | $\ell_\infty$ | 98.0$\pm$2.0 | 98.0$\pm$2.3 | 96.0$\pm$3.3 | 98.0$\pm$1.7 | 99.5$\pm$0.9 | 99.5$\pm$0.9 | 99.0$\pm$1.0 | 99.5$\pm$0.5 |
| | $\ell_2$ | 97.5$\pm$1.9 | 97.4$\pm$2.2 | 94.9$\pm$3.2 | 97.4$\pm$1.7 | 99.4$\pm$1.1 | 99.2$\pm$1.0 | 98.6$\pm$1.4 | 99.3$\pm$0.7 |
| | $\ell_1$ | 87.4$\pm$31.8 | 90.5$\pm$26.4 | 77.9$\pm$38.4 | 89.0$\pm$19.2 | 99.9$\pm$0.4 | 99.9$\pm$0.4 | 99.8$\pm$0.5 | 99.9$\pm$0.3 |

Here, $\eta > 0$ is the mutation step size and the mutation operator is applied to each individual with probability $p_m$. Then elite preservation retains the top-$k$ individuals for the next generation. The fundamental limitation of traditional crossover is that its discrete, element-wise mixing tends to break spatial and structural coherence of successful adversarial patterns. Randomly combining pixels or features from two strong parents can produce offspring that no longer fool the classifier. Moreover, uniform crossover is agnostic to the loss landscape between parents and may create children that lie in regions of low adversarial effectiveness. Finally, the element-wise nature restricts the search to certain combinations of parent features and can limit exploration of more structured adversarially effective paths between adversarial modes.

### 4.2. MoCo-EA Algorithm Overview

We propose MoCo-EA, which enhances traditional evolutionary algorithms by replacing the traditional discrete crossover operator with a geometry-aware Bézier crossover. The overall algorithm maintains a population of candidate perturbations and evolves them through selection, Bézier crossover, and mutation, following the general structure of EAs but innovating in its crossover mechanism. The Bézier crossover operator is detailed below in Algorithm 1. For the complete MoCo-EA procedure, please refer to Appendix B. An overview of the pipeline is shown in Figure 1.

The procedure begins by initializing a population of $N$ random perturbations inside the $\ell_p$ $\epsilon$-ball around the input. Each perturbation is evaluated with a fitness function based on its ability to cause misclassification. In each generation, parent pairs are selected from the population according to their fitness. The Bézier crossover operator then takes two parents, $\delta_1$ and $\delta_2$, and connects them with a quadratic Bézier curve parameterized by a control point $\boldsymbol{\delta}_c$. The control point $\boldsymbol{\delta}_c$ is optimized for a few gradient steps to maximize adversarial loss along sampled points on the path, with each point projected back to the $\epsilon$-ball to satisfy the perturbation constraint. After this optimization, new offspring are generated by sampling from different regions of the curve. Points closer to $\delta_1$ and $\delta_2$ are used to form distinct children, and among multiple samples the highest-fitness ones are chosen. Each selected offspring is projected back to the feasible set.

Mutation is applied to offspring with a certain probability. Elitism ensures that the top $k$ individuals from the current population are preserved. The new generation is then formed from the elites and the best offspring from crossover and mutation. This process repeats until either a successful adversarial perturbation is found or the maximum number of generations $G$ is reached.

Such trajectories instantiate adversarial mode connectivity and exhibit two important properties. First, connectivity: adversarial effectiveness is preserved along the path. Second, transferability: intermediate points on the curve often transfer better than the endpoints. Together, these properties position Bézier connectivity as an effective paradigm for structuring adversarial perturbations and provide a direct rationale for its use within an evolutionary search framework.

## 5. Experiments

### 5.1. Experimental Setup

**Datasets and models.** We evaluate on CIFAR-10 (Krizhevsky, 2009) with a ResNet-18 (He et al., 2016) classifier, and on ImageNet (Deng et al., 2009) with a ViT-Base/16 (Dosovitskiy et al., 2021) classifier. See Appendix C for dataset and model details.

*Table 2.* **Transferability along Bézier paths.** Results are reported on CIFAR-10 and ImageNet across three settings. "*Endp. Avg*" is the average success rate of the two endpoints, "*Path Succ.*" denotes the fraction of test images successfully attacked by at least one sampled point along the Bézier path, "*Imgs Resc.*" denotes the fraction of test images not attacked by endpoints but rescued by at least one path point, and "*Avg. pts.*" is the average number of successful path points per image, with 50 points sampled per curve. Values are mean ± standard deviation.

| Setting | Norm | CIFAR-10 | | | | ImageNet | | | |
|---|---|---|---|---|---|---|---|---|---|
| | | Endp. Avg | Path Succ. | Imgs Resc. | Avg. pts. | Endp. Avg | Path Succ. | Imgs Resc. | Avg. pts. |
| A | $\ell_\infty$ | $20.3 \pm 5.2$ | $34.7 \pm 5.2$ | $8.6 \pm 2.6$ | $12.7 \pm 2.5$ | $1.0 \pm 2.0$ | $3.5 \pm 5.7$ | $1.5 \pm 3.6$ | $0.8 \pm 1.5$ |
| | $\ell_2$ | $6.5 \pm 1.5$ | $9.4 \pm 2.2$ | $1.2 \pm 0.9$ | $3.6 \pm 0.7$ | $0.2 \pm 1.1$ | $0.5 \pm 2.2$ | $0.0 \pm 0.0$ | $0.0 \pm 0.1$ |
| | $\ell_1$ | $4.5 \pm 1.5$ | $11.2 \pm 2.3$ | $4.7 \pm 2.0$ | $3.4 \pm 0.8$ | $0.2 \pm 1.1$ | $2.0 \pm 4.0$ | $1.5 \pm 3.6$ | $0.7 \pm 1.4$ |
| B | $\ell_\infty$ | $20.4 \pm 3.6$ | $39.7 \pm 5.1$ | $11.8 \pm 3.2$ | $14.6 \pm 2.6$ | $0.5 \pm 1.5$ | $3.0 \pm 4.6$ | $2.0 \pm 4.0$ | $0.1 \pm 0.2$ |
| | $\ell_2$ | $6.5 \pm 1.3$ | $10.9 \pm 2.4$ | $1.8 \pm 1.7$ | $3.6 \pm 0.9$ | $0.0 \pm 0.0$ | $1.0 \pm 3.0$ | $1.0 \pm 3.0$ | $0.2 \pm 0.7$ |
| | $\ell_1$ | $4.8 \pm 1.4$ | $12.0 \pm 2.2$ | $4.5 \pm 1.7$ | $3.6 \pm 0.7$ | $0.0 \pm 0.0$ | $1.0 \pm 3.0$ | $1.0 \pm 3.0$ | $0.3 \pm 0.9$ |
| C | $\ell_\infty$ | $22.0 \pm 2.8$ | $38.2 \pm 4.3$ | $9.0 \pm 2.6$ | $13.9 \pm 2.0$ | $0.6 \pm 1.1$ | $2.8 \pm 3.3$ | $1.7 \pm 2.4$ | $0.7 \pm 1.1$ |
| | $\ell_2$ | $5.6 \pm 1.0$ | $9.9 \pm 1.9$ | $1.8 \pm 1.0$ | $2.9 \pm 0.8$ | $0.2 \pm 0.8$ | $1.2 \pm 2.2$ | $0.8 \pm 1.8$ | $0.2 \pm 0.6$ |
| | $\ell_1$ | $2.6 \pm 1.3$ | $8.4 \pm 2.9$ | $4.2 \pm 2.3$ | $2.4 \pm 0.7$ | $0.0 \pm 0.0$ | $0.8 \pm 1.8$ | $0.8 \pm 1.8$ | $0.1 \pm 0.4$ |

**Attack settings.** On CIFAR-10, we use $\ell_\infty$ with $\epsilon = 8/255$, $\ell_2$ with $\epsilon = 0.5$, and $\ell_1$ with $\epsilon = 10$. On ImageNet, we use $\ell_\infty$ with $\epsilon = 4/255$, $\ell_2$ with $\epsilon = 2$, and $\ell_1$ with $\epsilon = 75$. For generating adversarial endpoints, we employ Projected Gradient Descent (PGD) (Madry et al., 2018) with 40 iterations using step sizes $\alpha = \epsilon/4$ for $\ell_\infty$, $\alpha = \epsilon/5$ for $\ell_2$, and $\alpha = \epsilon/10$ for $\ell_1$. See Appendix C for the image selection protocols for Settings A/B/C.

**Bézier optimization.** The control point $\delta_c$ is optimized using the Adam optimizer (Kingma & Ba, 2015) with a learning rate of 0.01 for 30 iterations. We sample 20 random $t$ values per iteration during optimization, and evaluate the final curve on 50 evenly spaced $t$ values (excluding endpoints) to compute success rates. For MoCo-EA crossover, we use a reduced 5 iterations with 3 fixed sampling points $t \in \{0.25, 0.5, 0.75\}$ for efficiency.

### 5.2. Adversarial Mode Connectivity

**Connectivity analysis.** We test whether continuous paths between successful adversarial perturbations preserve attack effectiveness. We evaluate attack success along optimized Bézier paths with 25 data cases per setting. For each setting we (i) obtain two adversarial endpoints with PGD, (ii) build a quadratic Bézier curve $B(t; \delta_c)$ between them and optimize the control point $\delta_c$, and (iii) evaluate success by sampling $t \in [0.02, 0.98]$ and checking whether $f_\theta\big(x + \Pi_{\|\cdot\|_p \leq \epsilon}[B(t; \delta_c)]\big)$ is misclassified. Table 1 shows that optimizing the Bézier control point yields smooth adversarial paths that retain attack strength across intermediate points, supporting Bézier-based path construction and downstream uses (*e.g.*, Bézier crossover in MoCo-EA). We compare against linear interpolation and observe a substantial drop in connectivity, especially in Settings B and C. On ImageNet, ASR Both drops to about 12–37% under linear paths, whereas optimized Bézier paths in Table 1 remain

*Table 3.* **Effect of multi-image augmentation on CIFAR-10 under $\ell_\infty$.** "*Imp.*" is the difference between Path Succ. and Endp. Avg. "*Rescue Rate*" is the fraction of test images that failed at endpoints but succeeded along the path. "*Aux*" denotes the number of auxiliary images used. Values are mean ± standard deviation.

| Setting | Aux | Endp. Avg | Path Succ. | Imp. | Rescue Rate |
|---|---|---|---|---|---|
| A | 0 | $18.2 \pm 1.6$ | $32.4 \pm 4.8$ | $+14.2$ | $8.0 \pm 3.3$ |
| | 5 | $18.2 \pm 1.6$ | $44.4 \pm 3.6$ | $+26.2$ | $20.2 \pm 2.6$ |
| | 10 | $18.2 \pm 1.6$ | $50.0 \pm 2.7$ | $+31.8$ | $25.6 \pm 2.9$ |
| | 15 | $18.2 \pm 1.6$ | $56.4 \pm 2.6$ | $+38.2$ | $31.8 \pm 1.3$ |
| | 20 | $18.2 \pm 1.6$ | $57.4 \pm 4.5$ | $+39.2$ | $33.0 \pm 3.0$ |
| | 25 | $18.2 \pm 1.6$ | $58.2 \pm 5.2$ | $+40.0$ | $33.8 \pm 4.9$ |
| B | 0 | $22.2 \pm 3.7$ | $43.4 \pm 7.5$ | $+21.2$ | $13.8 \pm 5.2$ |
| | 5 | $22.2 \pm 3.7$ | $49.2 \pm 5.4$ | $+27.0$ | $19.8 \pm 2.5$ |
| | 10 | $22.2 \pm 3.7$ | $54.4 \pm 4.9$ | $+32.2$ | $24.6 \pm 2.7$ |
| | 15 | $22.2 \pm 3.7$ | $58.6 \pm 5.8$ | $+36.4$ | $28.8 \pm 4.8$ |
| | 20 | $22.2 \pm 3.7$ | $60.8 \pm 4.1$ | $+38.6$ | $31.0 \pm 3.5$ |
| | 25 | $22.2 \pm 3.7$ | $61.8 \pm 6.6$ | $+39.6$ | $32.0 \pm 5.2$ |
| C | 0 | $21.7 \pm 2.8$ | $41.4 \pm 3.8$ | $+19.7$ | $12.2 \pm 4.1$ |
| | 5 | $21.7 \pm 2.8$ | $43.2 \pm 5.2$ | $+21.5$ | $13.8 \pm 6.2$ |
| | 10 | $21.7 \pm 2.8$ | $46.6 \pm 3.8$ | $+24.9$ | $17.0 \pm 6.2$ |
| | 15 | $21.7 \pm 2.8$ | $44.6 \pm 0.8$ | $+22.9$ | $15.2 \pm 2.9$ |
| | 20 | $21.7 \pm 2.8$ | $44.8 \pm 1.6$ | $+23.1$ | $15.2 \pm 3.7$ |
| | 25 | $21.7 \pm 2.8$ | $51.8 \pm 1.9$ | $+30.1$ | $22.0 \pm 4.6$ |

near-perfect. This highlights that adversarial connectivity is not a trivial consequence of interpolation, but optimizing a curved path is important for preserving adversarial effectiveness (see Appendix D.1).

**Transferability analysis.** We further investigate whether adversarial connectivity improves transferability across different images and settings. Specifically, we compare endpoint-average transferability with connectivity-based path transferability across $\ell_\infty$, $\ell_2$, and $\ell_1$ norms. Evaluation is conducted on unseen images using curves optimized from training cases. On CIFAR-10, we use 25 training samples per setting. On ImageNet, we use 20 training samples for Settings

*Table 4.* **Convergence and sampling density on CIFAR-10 under $\ell_\infty$.** We report (a) convergence across training epochs with 100 sampled points per curve and (b) coverage per point under different sampling densities. All values are mean ± standard deviation.

(a) Convergence across epochs (100 points).

| Setting | Aux | 10 epochs | 20 epochs | 30 epochs | 40 epochs | 50 epochs |
|---|---|---|---|---|---|---|
| A | 0 | 29.8±5.1 | 32.6±4.6 | 33.6±5.7 | 34.4±4.9 | 35.2±4.1 |
| | 5 | 37.0±2.9 | 42.2±4.8 | 44.8±5.4 | 46.2±3.1 | 46.4±3.3 |
| | 10 | 37.2±3.1 | 45.2±4.8 | 48.8±2.9 | 51.6±2.4 | 53.2±3.4 |
| | 15 | 38.8±2.5 | 49.4±4.8 | 56.4±4.2 | 58.6±2.9 | 58.4±2.5 |
| | 20 | 41.0±2.6 | 52.6±3.9 | 57.0±2.3 | 58.8±3.0 | 60.2±2.6 |
| | 25 | 43.6±5.3 | 56.2±5.3 | 60.2±3.8 | 60.4±5.4 | 61.6±5.0 |
| B | 0 | 40.4±6.8 | 43.0±8.2 | 44.4±7.6 | 43.6±6.8 | 44.2±6.8 |
| | 5 | 46.0±5.3 | 48.4±6.3 | 50.4±7.1 | 50.2±6.6 | 51.0±6.7 |
| | 10 | 45.8±6.4 | 51.0±4.6 | 54.2±3.8 | 56.0±4.5 | 56.2±4.4 |
| | 15 | 47.6±6.5 | 55.8±4.8 | 57.8±4.7 | 59.8±4.7 | 60.6±3.4 |
| | 20 | 50.0±6.8 | 58.0±2.9 | 60.8±3.1 | 60.6±4.4 | 60.6±5.0 |
| | 25 | 49.8±7.1 | 57.0±7.1 | 60.0±6.5 | 60.4±5.2 | 61.2±5.6 |
| C | 0 | 37.2±4.8 | 39.2±3.7 | 40.2±3.8 | 40.8±4.1 | 40.6±3.5 |
| | 5 | 38.4±3.4 | 41.0±3.4 | 43.4±3.3 | 44.2±4.5 | 45.4±5.2 |
| | 10 | 39.0±1.4 | 44.4±4.2 | 45.0±2.3 | 46.2±4.9 | 46.8±3.3 |
| | 15 | 39.2±2.6 | 42.6±4.3 | 45.2±2.9 | 46.2±3.4 | 46.2±2.9 |
| | 20 | 39.6±2.9 | 45.0±3.3 | 46.4±2.2 | 46.0±2.3 | 46.0±3.0 |
| | 25 | 40.6±3.3 | 49.0±2.6 | 52.2±4.4 | 53.4±4.2 | 53.4±4.3 |

(b) Coverage under sampling densities.

| Setting | Aux | 50 points | 100 points |
|---|---|---|---|
| A | 0 | 26.0±2.4 | 26.2±2.5 |
| | 5 | 37.0±2.2 | 37.1±2.2 |
| | 10 | 44.5±3.9 | 44.7±4.0 |
| | 15 | 50.6±2.6 | 50.9±2.6 |
| | 20 | 51.5±3.8 | 51.8±3.7 |
| | 25 | 53.7±5.2 | 54.1±5.3 |
| B | 0 | 34.4±7.6 | 34.4±7.6 |
| | 5 | 41.0±6.7 | 41.2±6.7 |
| | 10 | 47.8±4.2 | 48.1±4.2 |
| | 15 | 52.2±3.3 | 52.5±3.4 |
| | 20 | 53.0±4.3 | 53.3±4.3 |
| | 25 | 53.9±5.0 | 54.2±5.1 |
| C | 0 | 29.7±4.1 | 29.8±4.1 |
| | 5 | 35.1±4.1 | 35.2±4.2 |
| | 10 | 36.7±5.2 | 36.8±5.3 |
| | 15 | 36.4±2.1 | 36.6±2.2 |
| | 20 | 36.9±4.5 | 37.0±4.6 |
| | 25 | 44.7±4.6 | 44.9±4.7 |

A and C, and 10 training samples for Setting B. Table 2 shows that connectivity-based paths consistently improve transferability across all norms and settings. This demonstrates that adversarial mode connectivity enables more robust and transferable perturbations, significantly enhancing the effectiveness of attacks beyond isolated adversarial examples. We hypothesize that the observed transferability gains arise because intermediate points along optimized paths frequently outperform the endpoints in transfer, indicating that the curve traverses flatter, more universal regions of the loss landscape. This may explain the observed improvements in reliability and cross-instance generalization.

**Multi-image augmentation analysis.** We study how adding $N$ auxiliary images when optimizing the Bézier control point affects transferability, varying auxiliary images $N \in \{0, 5, 10, 15, 20, 25\}$ with five repetitions. Table 3 shows that adding auxiliary images generally improves path success and rescue rate across all settings. Multi-image augmentation both regularizes the curve optimization and discovers more universal adversarial patterns that transfer to unseen images.

**Convergence and sampling density analysis.** For each auxiliary-image count, we evaluate how *coverage*, the percentage of test images that are successfully attacked by at least one sampled point along the Bézier path, changes as the number of epochs increases. We consider epochs $\{10, 20, 30, 40, 50\}$, each repeated 5 times. We also evaluate two sampling densities along each Bézier curve, using 50 or 100 sampled points on the curve, and under each density report *coverage per point*, defined as the average number

of images that a single sampled point successfully attacks. Table 4 shows that for a fixed auxiliary setting, increasing the number of optimization epochs generally improves coverage, and larger auxiliary sets achieve higher final coverage while typically requiring more epochs for the gains to fully materialize. Using more sampled points on each Bézier curve (100 vs. 50) yields slightly higher measured coverage per point, and the effect is stronger when more auxiliary images are available.

### 5.3. Mode Connectivity Evolutionary Attack

We first evaluate the MoCo-EA method against the traditional evolutionary algorithm baseline, focusing on key performance outcomes. The baseline follows a standard population-based pipeline, and MoCo-EA keeps this pipeline unchanged, differing only in the crossover step, where it replaces element-wise crossover with our geometry-aware Bézier crossover. Additional details are provided in Appendix C. This comparison allows us to assess how the connectivity and transferability advantages of Bézier paths translate into practical improvements for evolutionary adversarial attacks.

Table 5 summarizes the comparative performance of MoCo-EA and the traditional evolutionary algorithm baseline on CIFAR-10 and ImageNet under the $\ell_\infty$, $\ell_2$, and $\ell_1$ perturbation norms. MoCo-EA consistently surpasses the traditional EA across every performance dimension. It achieves near-perfect *success rates*, even under $\ell_2$ and $\ell_1$ norm constraints where the baseline often fails, while requiring only a handful of *generations* compared to the hundreds typi-

*Table 5.* **Comparison with the traditional EA baseline.** Results on CIFAR-10 and ImageNet with a population size of 30. "*Succ. rate*" denotes the percentage of successful attacks, "*Avg. gen.*" denotes the mean number of generations over successful attacks, "*Avg. queries*" denotes the mean number of model forward evaluations over all attempted samples, including those used during Bézier crossover. "*Avg. time*" denotes the mean runtime in seconds over all attempted samples, and "*Imp.*" denotes the corresponding improvement, i.e., percentage-point increase for success rate and percentage reduction for the other metrics. Averaged values are reported as mean ± standard deviation.

| Norm | Metric | CIFAR-10 | | | ImageNet | | |
|---|---|---|---|---|---|---|---|
| | | Traditional | MoCo-EA | Imp. | Traditional | MoCo-EA | Imp. |
| $\ell_\infty$ | Succ. rate | 93.3 | **100.0** | +6.7 | 83.3 | **100.0** | +16.7 |
| | Avg. gen. | $367.9 \pm 233.2$ | $\mathbf{1.7 \pm 1.1}$ | ↓99.5% | $456.8 \pm 309.0$ | $\mathbf{1.0 \pm 0.0}$ | ↓99.8% |
| | Avg. queries | $12329 \pm 8247$ | $\mathbf{628 \pm 367}$ | ↓94.9% | $16446 \pm 10408$ | $\mathbf{375 \pm 0}$ | ↓97.7% |
| | Avg. time | $29.44 \pm 19.72$ | $\mathbf{6.08 \pm 3.73}$ | ↓79.3% | $95.14 \pm 60.22$ | $\mathbf{5.05 \pm 0.04}$ | ↓94.7% |
| $\ell_2$ | Succ. rate | 6.7 | **100.0** | +93.3 | 13.3 | **100.0** | +86.7 |
| | Avg. gen. | $25.0 \pm 19.0$ | $\mathbf{1.4 \pm 1.6}$ | ↓94.4% | $24.8 \pm 9.8$ | $\mathbf{1.0 \pm 0.0}$ | ↓96.0% |
| | Avg. queries | $28052 \pm 7290$ | $\mathbf{513 \pm 561}$ | ↓98.2% | $26103 \pm 9936$ | $\mathbf{375 \pm 0}$ | ↓98.6% |
| | Avg. time | $67.94 \pm 17.67$ | $\mathbf{4.97 \pm 5.65}$ | ↓92.7% | $152.15 \pm 58.07$ | $\mathbf{5.01 \pm 0.02}$ | ↓96.7% |
| $\ell_1$ | Succ. rate | 56.7 | **100.0** | +43.3 | 33.3 | **100.0** | +66.7 |
| | Avg. gen. | $55.8 \pm 196.9$ | $\mathbf{1.0 \pm 0.5}$ | ↓98.2% | $13.3 \pm 22.6$ | $\mathbf{0.9 \pm 0.3}$ | ↓93.2% |
| | Avg. queries | $13966 \pm 14709$ | $\mathbf{375 \pm 178}$ | ↓97.3% | $20143 \pm 13945$ | $\mathbf{340 \pm 104}$ | ↓98.3% |
| | Avg. time | $34.82 \pm 36.64$ | $\mathbf{3.74 \pm 1.87}$ | ↓89.3% | $118.19 \pm 81.81$ | $\mathbf{4.61 \pm 1.48}$ | ↓96.1% |

*Table 6.* **Comparison with gradient-based attacks.** Results are reported on CIFAR-10 under the $\ell_\infty$ norm in two settings: (a) a robustly trained model and (b) obfuscated gradients. For each setting, ASR is computed over the test images correctly classified before attack. We evaluate 8,703 images for the robust-model setting and 7,510 images for the obfuscated-gradient setting. We report attack success rates (ASR, %).

| Setting | PGD | MI-FGSM | AA | AAA | MoCo-EA |
|---|---|---|---|---|---|
| Robust model | 39.6 | 38.7 | 40.3 | 39.1 | **40.5** |
| Obfuscated gradients | 13.5 | 13.3 | 0.0 | 12.8 | **53.8** |

cally needed by the baseline. This efficiency translates into dramatically fewer *queries*, as offspring are sampled along optimized adversarially effective paths rather than through costly trial-and-error exploration. Consequently, the *runtime* improvements follow naturally, with MoCo-EA completing attacks substantially faster than its counterpart. Together, these results confirm that incorporating Bézier connectivity into evolutionary search yields uniformly superior performance by transforming recombination from random mixing into geometry-aware exploration of connected adversarial manifolds. Appendix D.3 provides a metric-wise analysis of these results. In addition, Appendix D.4 shows that population size primarily influences the efficiency of MoCo-EA, while its reliability remains consistently high across different population settings.

**Comparison with gradient-based attacks.** To compare MoCo-EA with gradient-based adversarial attacks, we consider two experimental settings on CIFAR-10. First, we evaluate attack success rates on a robustly trained model using the adversarially trained CIFAR-10 ResNet-50 checkpoint released by Engstrom et al. (2019). We report *attack success*

*rates* (ASR) over test images correctly classified before attack. For this robust model setting, we evaluate all 8,703 correctly classified test images using PGD (Madry et al., 2018), MI-FGSM (Dong et al., 2018), AutoAttack (AA) (Croce & Hein, 2020), Adaptive AutoAttack (AAA) (Liu et al., 2022), and our MoCo-EA. As shown in Table 6, MoCo-EA achieves the highest ASR among the compared methods, although the gain over AA is modest. This suggests that MoCo-EA is competitive with strong gradient-based attacks in this setting. Second, we evaluate performance under an obfuscated-gradient setting (Athalye et al., 2018) on CIFAR-10. Specifically, we apply an additional quantization step to the input image, `torch.round(x × 5)/5`, which makes gradients vanish in most regions and therefore breaks conventional gradient-based attacks. For this setting, we evaluate all 7,510 correctly classified test images. In this scenario, gradient-based attacks become unreliable or ineffective, whereas MoCo-EA leverages population-based search and is therefore less dependent on a single reliable gradient trajectory. As shown in Table 6, MoCo-EA substantially outperforms the gradient-based baselines under obfuscated gradients. MoCo-EA is not intended to replace gradient-based attacks, but rather to highlight its distinct role in understanding adversarial geometry and in handling cases where gradients are unreliable. Unlike conventional methods that follow a single optimization trajectory, MoCo-EA evolves an entire population of perturbations, enabling broader exploration of the adversarial landscape. We also report the ImageNet results in Appendix D.2, where MoCo-EA achieves the highest ASR in both settings.

## 6. Computational Cost Analysis

The primary efficiency gain of MoCo-EA does not come from reducing the per-generation cost, but from faster convergence. Bézier crossover introduces additional forward and backward computations compared with the baseline EA. However, as shown in Table 5, this additional per-generation overhead is offset in practice by the much smaller number of generations and total forward evaluations required for success. Let $C_{\text{fw}}$ and $C_{\text{bw}}$ denote the cost of one forward and one single-point backward-equivalent computation, respectively, and let $N$ be the population size. For the baseline EA, each generation evaluates $N$ individuals, giving a dominant per-generation cost of $O(NC_{\text{fw}})$, up to lower-order element-wise crossover and mutation operations. For MoCo-EA, each non-terminal generation with $\lfloor N/2 \rfloor$ parent pairs requires $N + (\tau s + k)\left\lfloor \frac{N}{2} \right\rfloor$ forward evaluations and $\tau s \left\lfloor \frac{N}{2} \right\rfloor$ single-point backward-equivalent computations, where $\tau$ is the number of control-point optimization steps, $s$ is the number of sampled curve points per step, and $k$ is the number of candidate points evaluated during offspring selection. Thus, the dominant per-generation cost of MoCo-EA is $\left(N + \left\lfloor \frac{N}{2} \right\rfloor (\tau s + k)\right) C_{\text{fw}} + \left\lfloor \frac{N}{2} \right\rfloor \tau s C_{\text{bw}}$. With our default setting ($N = 30, \tau = 5, s = 3, k = 6$), this corresponds to $345C_{\text{fw}} + 225C_{\text{bw}}$ in single-point backward-equivalent cost, compared with $30C_{\text{fw}}$ for the baseline EA. In the implementation, this corresponds to $\tau \lfloor N/2 \rfloor = 75$ backward calls per non-terminal generation, each over $s = 3$ sampled curve points. Therefore, MoCo-EA's computational advantage comes from requiring far fewer generations to find successful adversarial perturbations. Empirically, this results in a 94–99% reduction in total forward evaluations and lower end-to-end runtime than the baseline EA, as reported in Table 5. For reference, PGD-40 uses roughly $40C_{\text{fw}} + 40C_{\text{bw}}$. MoCo-EA instead uses geometry-aware evolutionary search, which can be beneficial in settings such as obfuscated gradients, where gradient-based attacks can be less effective (Table 6).

## 7. Conclusion

We introduced MoCo-EA, a novel approach that rethinks crossover operations in evolutionary adversarial attacks through continuous path optimization. By exploiting the mode connectivity property of adversarial perturbations, we demonstrated that successful attacks do not lie at isolated points, but rather on connected manifolds that can be traversed via optimized Bézier curves. Our experiments revealed that intermediate points along these paths exhibit higher transferability than endpoints, while replacing discrete genetic crossover with continuous Bézier interpolation yields significant improvements in both efficiency and effectiveness, achieving near-perfect success across perturbation norms and substantially reducing the number of generations,

queries, and runtime. Beyond immediate benefits for adversarial attack generation, our findings highlight broader implications for understanding the geometric structure of adversarial space and suggest that defenses may need to consider the continuous nature of adversarial manifolds. Future work includes higher-order Bézier curves for more complex path optimization, defensive applications of adversarial connectivity, and extensions to other domains where evolutionary algorithms are applied.

### Reproducibility Statement

To ensure the reproducibility of our experimental results, we provide the source code at: `https://github.com/TIML-Group/MoCo-EA`.

### Acknowledgments

This work was supported in part by the National Science Foundation under grants IIS-2246157, FMitF-2319243, and the Department of Energy under grant DE-CR0000042. The project was also supported by computational resources provided by the NSF ACCESS and Argonne National Lab.

### Impact Statement

This work advances research on adversarial attacks by revealing the geometric structure of adversarial perturbation spaces and proposing a more efficient evolutionary attack framework. While adversarial attack methods can potentially be misused, our goal is to support the development and evaluation of more robust and secure machine learning systems by improving the understanding of adversarial vulnerabilities. The societal and ethical implications of this work are consistent with those commonly associated with adversarial machine learning research, and we do not identify additional risks beyond those already well studied.

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

# Appendix

This appendix includes the following materials: 1) Geometric Interpretation of Adversarial Mode Connectivity in ReLU Networks (Appendix A), 2) MoCo-EA Algorithm (Appendix B), 3) Experimental Setup Details (Appendix C), and 4) Additional Results and Analysis (Appendix D).

## A. Geometric Interpretation of Adversarial Mode Connectivity in ReLU Networks

We provide a geometric intuition, rather than a formal proof, for why adversarial perturbations found by different optimization runs can be connected by a smooth path that preserves adversarial behavior. In a ReLU network, each activation pattern defines a convex polytope in the input space, and the network is affine within each such activation region. Therefore, if two adversarial perturbations lie in the same activation region and the connecting path remains in the misclassified portion of that region, connectivity follows directly from convexity.

The more relevant case is when the two perturbations lie in different activation regions. Adjacent ReLU regions share a facet, and the network output is continuous across this facet. Thus, if a point on the shared facet is misclassified and is not exactly on the decision boundary, nearby points on both sides of the facet will also tend to remain misclassified. Repeating this local connection across adjacent regions can plausibly form extended connected components of the misclassification set

$$\mathcal{A} = \{\delta \mid f(x+\delta) \neq y, \ \|\delta\|_p \leq \epsilon\}. \tag{A1}$$

This suggests why adversarial regions may extend across many activation regions rather than appearing only as isolated pockets.

Under this interpretation, optimizing the Bézier control point can be viewed as bending the path away from correctly classified regions and toward adversarial corridors spanning multiple activation regions. This offers one possible intuition for the empirical path connectivity observed in Table 1. Extending this explanation beyond ReLU networks remains future work.

## B. MoCo-EA Algorithm

Complete pseudocode for MoCo-EA is provided in Algorithm B1.

**InitializePopulation.** Given an input image $\mathbf{x}$, a budget $\epsilon$, and population size $N$, this routine samples $N$ initial perturbations $\{\boldsymbol{\delta}_i\}_{i=1}^{N}$ either randomly or using PGD, inside the feasible $\ell_p$-ball, i.e., $\|\boldsymbol{\delta}_i\|_p \leq \epsilon$, and returns the set $P$.

**EvaluateFitness.** For each candidate $\delta \in P$, fitness is evaluated using the model's confidence in the true label. Successful perturbations are ranked above unsuccessful ones, and among candidates with the same success status, lower confidence in the true label indicates higher fitness. The routine returns a vector $F$ of fitness scores aligned with $P$.

**elite $\cup$ SelectBest.** Elitism first preserves the top-$k$ individuals from the current population $P$ according to $F$ as the set $\text{elite} = \text{SelectTop}(P, k)$. Then, among the newly generated offspring, $\text{SelectBest}(\cdot, N - k)$ picks the highest-fitness $(N - k)$ candidates. Their union $\text{elite} \cup \text{SelectBest}(\cdot)$ forms the next generation of size $N$ while guaranteeing that the strongest current solutions are never discarded.

**Remark.** The crossover operator used here is the geometry-aware Bézier subroutine in Algorithm 1. Section 4.2 provides the algorithmic overview.

## C. Experimental Setup Details

**CIFAR-10 and ResNet-18 details.** CIFAR-10 contains 50,000 training and 10,000 test images across 10 classes (Krizhevsky, 2009). We use a ResNet-18 (He et al., 2016) adapted for CIFAR-10 by replacing the initial $7 \times 7$ convolution with a $3 \times 3$ kernel (stride $= 1$, padding $= 1$) and removing the max-pooling layer. The model is trained for 200 epochs using SGD with momentum 0.9, weight decay $5 \times 10^{-4}$, and a multi-step learning-rate schedule (initial lr $= 0.1$, decayed by $10 \times$ at epochs 60, 120, 160), achieving $95.1\%$ clean test accuracy.

**ImageNet and ViT-Base/16 details.** For ImageNet (Deng et al., 2009), we evaluate on the standard validation set (50,000 images, 1,000 classes). We adopt a Vision Transformer (ViT-Base, patch size $16 \times 16$) (Dosovitskiy et al., 2021) pretrained on ImageNet. Preprocessing follows the common pipeline: resize the shorter side to 256, center crop to $224 \times 224$, and

---

**Algorithm B1** Mode Connectivity Evolutionary Attack (MoCo-EA)

---

**Input:** Image $\mathbf{x}$, label $y$, model $f_{\boldsymbol{\theta}}$, max generations $G$
**Parameter:** population size $N$, elite size $k$, mutation rate $p_m$, mutation std $\sigma$
**Output:** Adversarial perturbation $\boldsymbol{\delta}^*$
$P \leftarrow \text{InitializePopulation}(N, \epsilon)$
$\boldsymbol{\delta}^* \leftarrow \text{null}$
**for** $g = 1$ **to** $G$ **do**
    $F \leftarrow \text{EvaluateFitness}(P, \mathbf{x}, y, f_{\boldsymbol{\theta}})$
    **if** $\max(F) > \text{fitness}(\boldsymbol{\delta}^*)$ **then**
        $\boldsymbol{\delta}^* \leftarrow \arg\max_{\boldsymbol{\delta} \in P} \text{fitness}(\boldsymbol{\delta})$
    **end if**
    **if** $\text{IsSuccessful}(\boldsymbol{\delta}^*)$ **then**
        **return** $\boldsymbol{\delta}^*$
    **end if**
    $parents \leftarrow \text{TournamentSelection}(P, F)$
    $offspring \leftarrow \varnothing$
    **for each** $(p_1, p_2)$ **in** $parents$ **do**
        $(c_1, c_2) \leftarrow \text{BezierCrossover}(p_1, p_2, \mathbf{x}, y, f_{\boldsymbol{\theta}})$
        $offspring \leftarrow offspring \cup \{\text{Mutate}(c_1, p_m, \sigma), \text{Mutate}(c_2, p_m, \sigma)\}$
    **end for**
    $elite \leftarrow \text{SelectTop}(P, k)$
    $P \leftarrow elite \cup \text{SelectBest}(offspring, N - k)$
**end for**
**return** $\boldsymbol{\delta}^*$

---

normalize with the pretrained ViT statistics (mean $= 0.5$, std $= 0.5$). The pretrained ViT achieves $84.4\%$ top-1 accuracy on the validation set.

**Image selection protocol for Settings A/B/C.** The connectivity scenarios are fixed across datasets. On CIFAR-10, Setting A uses a single cat image, Setting B uses two cat images from the same class, and Setting C pairs a cat image with a dog image. On ImageNet, the same structure is applied with Egyptian cat images for Settings A and B and with an Egyptian cat image paired with a Labrador retriever image for Setting C.

**Baseline evolutionary algorithm details.** The baseline evolutionary attack follows a standard population-based procedure (Alzantot et al., 2019). It maintains a population of $N$ perturbations, where 30 is the default, and iteratively evolves them under the same $\ell_p$-norm ball. Each perturbation is initialized within the $\ell_p$ ball of radius $\epsilon$. For each candidate $\delta$, fitness is evaluated using the model's confidence in the true label. Successful perturbations are ranked above unsuccessful ones, and among candidates with the same success status, lower confidence in the true label indicates higher fitness. At every iteration, we preserve the top $K = 5$ elite individuals, while the remaining candidates for reproduction are selected via tournament selection with size $k = 3$. New offspring are generated using uniform crossover with probability $p = 0.5$, where each pixel is independently inherited from one of the two parents. After crossover, Gaussian mutation with standard deviation $0.02\epsilon$ is applied with probability 0.2, and the resulting perturbations are projected back onto the $\ell_p$ ball to satisfy the norm constraint. The next generation is formed by combining the preserved elites with the highest-fitness offspring. This iterative process is repeated for a maximum of $T = 1000$ iterations or until the model misclassifies the image.

# D. Additional Results and Analysis

### D.1. Linear Interpolation vs. Bézier Connectivity

We evaluate whether linear interpolation preserves connectivity on ImageNet using the same protocol as in Table 1. Given two adversarial endpoints $\delta_1$ and $\delta_2$, we construct a linear path $\delta_t = (1 - t)\delta_1 + t\delta_2$ and evaluate attack success along the path. As shown in Table D1, connectivity drops substantially in Settings B and C, indicating that linear interpolation does not reliably preserve adversarial effectiveness, especially in multi-image and cross-class settings. In contrast, Table 1 shows that optimized Bézier paths maintain near-perfect connectivity across all settings, demonstrating that adversarial

*Table D1.* **Connectivity under linear interpolation.** Results are reported on ImageNet across three settings with 25 data cases each (single images in Setting A and image pairs in Settings B and C). "*ASR1*" and "*ASR2*" denote the fraction of interior points that successfully attack the first and second image, respectively, when defined. "*ASR Both*" denotes the fraction of interior points that successfully attack both images simultaneously. In Setting A, it corresponds to the path success rate for the single image. "*ASR Avg.*" denotes the average of ASR1 and ASR2 when defined, and is set equal to "*ASR Both*" in Setting A. Attacks are evaluated along linear paths connecting the endpoints. All values are reported as mean $\pm$ standard deviation.

| Setting | Norm | ASR1 | ASR2 | ASR Both | ASR Avg. |
|---|---|---|---|---|---|
| A | $\ell_\infty$ | N/A | N/A | $100.0 \pm 0.0$ | $100.0 \pm 0.0$ |
| | $\ell_2$ | N/A | N/A | $100.0 \pm 0.0$ | $100.0 \pm 0.0$ |
| | $\ell_1$ | N/A | N/A | $97.6 \pm 11.8$ | $97.6 \pm 11.8$ |
| B | $\ell_\infty$ | $66.9 \pm 14.8$ | $68.0 \pm 16.4$ | $35.0 \pm 19.4$ | $67.4 \pm 9.8$ |
| | $\ell_2$ | $69.0 \pm 16.9$ | $68.2 \pm 18.9$ | $37.4 \pm 21.3$ | $68.6 \pm 10.9$ |
| | $\ell_1$ | $55.2 \pm 33.4$ | $63.9 \pm 28.8$ | $29.0 \pm 30.0$ | $59.6 \pm 21.6$ |
| C | $\ell_\infty$ | $67.0 \pm 17.4$ | $66.5 \pm 12.3$ | $33.5 \pm 20.2$ | $66.8 \pm 10.1$ |
| | $\ell_2$ | $67.2 \pm 17.1$ | $67.3 \pm 13.1$ | $35.1 \pm 18.8$ | $67.2 \pm 10.1$ |
| | $\ell_1$ | $51.1 \pm 33.0$ | $42.8 \pm 27.7$ | $11.8 \pm 18.0$ | $47.0 \pm 18.3$ |

*Table D2.* **Additional comparison with gradient-based attacks on ImageNet.** Results are reported under the $\ell_\infty$ norm in two settings: (a) an adversarially trained ImageNet model and (b) an obfuscated gradient setting using input quantization. All methods are evaluated on the same 100 randomly sampled ImageNet validation images in each setting. We report attack success rates (ASR, %).

| Setting | PGD | MI-FGSM | AA | AAA | MoCo-EA |
|---|---|---|---|---|---|
| Robust model | 38 | 38 | 38 | 43 | **47** |
| Obfuscated gradients | 17 | 17 | 17 | 16 | **32** |

connectivity is not a trivial consequence of interpolation and suggesting that optimized curved paths are important for preserving adversarial effectiveness.

### D.2. Additional ImageNet Results for Comparison with Gradient-based Attacks

We provide the ImageNet results of the gradient-based attack comparison in Table D2. The evaluation follows the same $\ell_\infty$ setting and reports attack success rates (ASR, %). We consider two ImageNet settings: a robust model and an obfuscated gradient setting. For the robust-model setting, we use the adversarially trained ResNet-50 checkpoint (Engstrom et al., 2019). For the obfuscated gradient setting, we use a standard ImageNet model with the same input quantization defense, `torch.round(x × 5)/5`, as in Table 6. All methods are evaluated on the same 100 randomly sampled ImageNet validation images in each setting.

As shown in Table D2, MoCo-EA achieves the highest ASR among the compared methods in both sampled ImageNet settings. These results complement the CIFAR-10 comparison in Table 6 and suggest that the observed trend also holds on ImageNet.

### D.3. Performance Metric Analysis

We provide a detailed examination of Table 5, analyzing performance metrics (success rate, generations, queries, and runtime) to better understand the improvements of MoCo-EA.

**Success rate.** MoCo-EA achieves consistently higher success rates than the traditional EA across all datasets and perturbation norms. While the baseline often struggles particularly under the $\ell_2$ and $\ell_1$ constraints, MoCo-EA attains near-perfect attack success across all tested settings. This improvement highlights the practical benefits of integrating Bézier connectivity. By ensuring that offspring perturbations lie on adversarially effective paths between adversarial modes, MoCo-EA preserves adversarial validity throughout the evolutionary process. The consistently superior success rates therefore demonstrate that connectivity and transferability properties observed in earlier analyses directly translate into more reliable attack generation within the evolutionary framework.

**Average generations.** MoCo-EA converges within only a few generations, in stark contrast to the baseline EA, which often requires hundreds of iterations to identify effective adversarial perturbations. This sharp reduction in generational

*Table D3.* **Effect of varying population size under $\ell_\infty$, $\ell_2$, and $\ell_1$ on CIFAR-10.** "*Succ. rate*" denotes success rate (%), "*Avg. gen.*" the average number of generations to success (successful cases only), "*Avg. queries*" the average number of queries, and "*Avg. time*" the average runtime in seconds. Results are reported for population sizes 15, 30, and 45, with each cell showing Traditional / MoCo-EA.

| Norm | Metric | 15 | 30 | 45 |
| --- | --- | --- | --- | --- |
| | | Traditional / MoCo-EA | Traditional / MoCo-EA | Traditional / MoCo-EA |
| $\ell_\infty$ | Succ. rate | 76.7 / **100.0** | 93.3 / **100.0** | 96.7 / **100.0** |
| | Avg. gen. | 487.8±221.2 / **1.9±1.3** | 367.9±233.2 / **1.7±1.1** | 315.9±231.0 / **1.7±0.9** |
| | Avg. queries | 9122±4354 / **317±208** | 12329±8247 / **628±367** | 15286±11614 / **890±460** |
| | Avg. time | 21.63±10.24 / **2.96±1.78** | 29.44±19.72 / **6.08±3.73** | 36.36±27.60 / **8.44±4.53** |
| $\ell_2$ | Succ. rate | 3.3 / **100.0** | 6.7 / **100.0** | 10.0 / **100.0** |
| | Avg. gen. | 7.0±0.0 / **2.4±6.6** | 25.0±19.0 / **1.4±1.6** | 20.7±11.1 / **1.7±3.4** |
| | Avg. queries | 14504±2671 / **398±1074** | 28052±7290 / **513±561** | 40598±13208 / **907±1728** |
| | Avg. time | 35.05±6.46 / **3.91±10.77** | 67.94±17.67 / **4.97±5.65** | 97.78±31.80 / **8.88±17.58** |
| $\ell_1$ | Succ. rate | 40.0 / **100.0** | 56.7 / **100.0** | 70.0 / **100.0** |
| | Avg. gen. | 96.8±170.0 / **1.0±0.5** | 55.8±196.9 / **1.0±0.5** | 8.2±7.3 / **1.0±0.4** |
| | Avg. queries | 9587±6823 / **177±84** | 13966±14709 / **375±178** | 13790±20434 / **535±206** |
| | Avg. time | 23.95±17.03 / **1.75±0.88** | 34.82±36.64 / **3.74±1.87** | 34.52±51.10 / **5.31±2.18** |

cost illustrates the efficiency of the Bézier crossover operator. Instead of relying on random recombination that frequently disrupts adversarial structure, offspring are sampled along optimized trajectories that reliably preserve attack validity. As a result, MoCo-EA transforms the evolutionary process from a slow, trial-and-error search into a rapid and directed exploration of adversarial space.

**Average queries.** MoCo-EA requires dramatically fewer model queries compared to the traditional EA baseline. By exploiting Bézier connectivity, the search process is guided toward regions of perturbation space that are already adversarially valid, thereby reducing the need for extensive query-based exploration. This efficiency gain is particularly significant under all norm constraints, where traditional EA must rely on large query budgets to locate viable perturbations. The reduction in query complexity underscores the practical value of geometry-aware crossover, making MoCo-EA more practical when model forward evaluations are limited or costly.

**Average runtime.** The improvements in average runtime follow naturally from the substantial reductions in generations and queries. Since MoCo-EA converges quickly and requires far fewer interactions with the target model, its wall-clock time is consistently lower than that of the traditional EA baseline. This outcome is therefore an expected consequence of the algorithm's efficiency gains, further confirming the practicality of integrating Bézier connectivity into evolutionary attacks.

### D.4. Ablation Study: Effect of Population Size

We further analyze the effect of population size (15/30/45) under $\ell_\infty$, $\ell_2$, and $\ell_1$ norms on CIFAR-10, as summarized in Table D3. Compared to Table 5, which reports results at a fixed population size, this ablation reveals how varying the population influences performance across the three norms. Three observations emerge:

**(i) Success rate vs. population size.** For the traditional EA, increasing the population improves success rate but leaves it far from reliable under $\ell_2$ (*e.g.*, 3.3% → 10.0% as population grows from 15 to 45), and still below 100% under $\ell_\infty/\ell_1$. In contrast, MoCo-EA attains 100% success across all tested populations and norms, indicating that geometry-aware crossover removes the method's reliance on large populations to achieve reliability.

**(ii) Generational cost and its interpretability.** For the traditional EA, average generations decrease as population grows under $\ell_\infty$ and $\ell_1$ (*e.g.*, 487.8 → 315.9 and 96.8 → 8.2), consistent with diversity aiding convergence. However, under $\ell_2$ the trend is inconsistent (7.0 → 25.0 → 20.7). This inconsistency is expected because the metric is computed only over successful attacks. When success is rare, the estimate becomes sample-size sensitive and is not representative of the algorithm's typical behavior. MoCo-EA, by contrast, converges in about one to two generations across all populations and norms, with small dispersion, reflecting a search guided along paths that preserve attack validity.

**(iii) Query/runtime scaling with population.** For the traditional EA, average queries and wall-clock time increase with population across all norms (*e.g.*, $\ell_\infty$ queries 9122 → 15286; $\ell_2$ time 35.05s → 97.78s), because per-generation evaluation cost grows with the number of individuals and the generational reduction is insufficient to offset this. MoCo-EA exhibits the

same linear-like scaling in queries/time with population (*e.g.*, $\ell_\infty$ queries $317 \rightarrow 890$), but since it typically converges in one or two generations, the absolute cost remains low (single-digit seconds), and the success rate does not benefit from larger populations. Consequently, smaller populations (*e.g.*, 15) already deliver the desired reliability and minimize query/time budgets.

Table 5 demonstrates MoCo-EA's advantage at a population size of 30. The ablation in Table D3 further shows that this advantage is robust across population sizes. MoCo-EA maintains $100\%$ success and near-constant generational cost for $15 \leq$ population $\leq 45$, while its query/time overhead grows approximately with population size. Traditional EA, in contrast, exhibits a classical exploration–efficiency trade-off. Larger populations yield somewhat higher success rates and fewer generations under $\ell_\infty/\ell_1$, but at the price of substantially higher queries and time, and still fail to produce reliable success under $\ell_2$.

Population size acts as a critical efficiency knob rather than a reliability enabler for MoCo-EA. Since geometry-aware crossover already enables connected, adversarially effective exploration, increasing the population provides no success-rate gain and only inflates queries/runtime. Hence, MoCo-EA's reliability is population-insensitive on CIFAR-10 across all tested norms, and its most resource-efficient regime is attained at smaller populations. For the traditional EA, larger populations partially compensate for unguided recombination by improving success and reducing generations under $\ell_\infty/\ell_1$, but they remain inefficient and ineffective under $\ell_2$, underscoring the central role of the geometry-aware prior introduced by Bézier connectivity.

