# OpenReview forum: "MoCo-EA: Exploiting Adversarial Mode Connectivity for Efficient Evolutionary Attacks"
_ICML.cc/2026/Conference — ICML 2026 regular_

### Official Review · Reviewer_7mWj · 2026-02-28

**Soundness:** 3
**Presentation:** 3
**Significance:** 4
**Originality:** 4
**Overall Recommendation:** 5
**Confidence:** 4

**Summary:**

This paper give a novel manifold insight for adversarial perturbations, which exploits successful adversarial perturbations found by different PGD runs can be connected by continuous paths along which adversarial effectiveness is preserved. The authors replace the traditional element-wise crossover in evolutionary attacks with a Bézier curve-based crossover operator. The resulting method achieves near-perfect attack success rates on CIFAR-10 and ImageNet, while dramatically reducing the number of generations, queries, and wall-clock time compared to the traditional EA baseline.

**Compliance With Llm Reviewing Policy:**

Affirmed.

**Final Justification:**

I think it's a solid work for adversarial perturbations. Even it also has some concers such as geometric intuition, it is still a solid work.

**Key Questions For Authors:**

1.Please report the ASR along the path when using simple linear interpolation without optimizing any control point. This is the most direct experiment to disentangle the contribution of the Bézier geometric structure from that of the optimization process itself. If linear interpolation already yields high connectivity, the necessity of the Bézier formulation becomes questionable.

2.The paper claims transferability "increases monotonically" with more auxiliary images, yet Table 3 shows a drop from 46.6% to 44.6% in Setting C at aux=15. Could the authors reconcile this inconsistency with their monotonicity claim?

**Limitations:**

Although the discovery of adversarial mode connectivity has natural implications for defense design, the paper only superficially mentions this in the conclusion without providing concrete analysis or preliminary experiments.

**Strengths And Weaknesses:**

Strengths:

1. Novel geometric perspective. Transferring mode connectivity from parameter space to adversarial input space is a creative and well-motivated idea that opens new directions for understanding adversarial geometry.

2. Strong and consistent empirical results. MoCo-EA achieves 100% success across all tested norms and datasets, with dramatic reductions in generations, queries, and runtime. The gains are especially notable under l2 and l1, where the baseline EA largely fails.

3. Robustness under obfuscated gradients. Table 6 shows meaningful improvement over gradient-based baselines in the input quantization setting, suggesting practical utility beyond standard white-box scenarios.

Weaknesses:

1. Table 1's near-perfect connectivity rates are achieved via active optimization of $\delta_c$. The paper never reports connectivity under simple linear interpolation without any control point optimization. Without this, it is impossible to attribute the improvement to the Bézier geometric structure rather than the optimization process itself.

2. The geometric intuition provided in Appendix A.1 is explicitly informal and restricted to ReLU networks. Since the ImageNet experiments use ViT-Base/16, which relies on attention mechanisms rather than ReLU activations, the theoretical justification for adversarial mode connectivity does not cover the primary experimental setting.

3. The paper claims transferability increases monotonically with the number of auxiliary images, yet Table 3 shows a clear drop in Setting C (46.6% to 44.6% at aux=10 to 15), directly contradicting this claim. This inconsistency is not acknowledged in the main text.

---

> ### Author Rebuttal · Authors · 2026-03-31
>
> We thank the reviewer for the valuable feedback and for recognizing the novelty, strong empirical results, and practical utility beyond standard white-box settings. We address the reviewer’s suggestions on the added value of the Bézier formulation, the scope of the geometric intuition, and potential overstatements below.
>
> ### **W1 & Q1) Evaluate connectivity under linear interpolation to isolate the effect of the Bézier formulation.**
>
> We use simple linear interpolation between two adversarial perturbations, $\delta_t = (1 - t)\delta_1 + t\delta_2$, and evaluate attack success along the resulting path on ImageNet. The results are reported using the same metrics as in Table 1.
>
> | Setting | Norm | ASR1 | ASR2 | ASR Both | ASR Avg |
> |--------|------|------|------|----------|---------|
> | A | ℓ∞ | N/A | N/A | 100.0 ± 0.0 | 100.0 ± 0.0 |
> | A | ℓ₂ | N/A | N/A | 100.0 ± 0.0 | 100.0 ± 0.0 |
> | A | ℓ₁ | N/A | N/A | 97.6 ± 11.8 | 97.6 ± 11.8 |
> | B | ℓ∞ | 66.9 ± 14.8 | 68.0 ± 16.4 | 35.0 ± 19.4 | 67.4 ± 9.8 |
> | B | ℓ₂ | 69.0 ± 16.9 | 68.2 ± 18.9 | 37.4 ± 21.3 | 68.6 ± 10.9 |
> | B | ℓ₁ | 55.2 ± 33.4 | 63.9 ± 28.8 | 29.0 ± 30.0 | 59.6 ± 21.6 |
> | C | ℓ∞ | 67.0 ± 17.4 | 66.5 ± 12.3 | 33.5 ± 20.2 | 66.8 ± 10.1 |
> | C | ℓ₂ | 67.2 ± 17.1 | 67.3 ± 13.1 | 35.1 ± 18.8 | 67.2 ± 10.1 |
> | C | ℓ₁ | 51.1 ± 33.0 | 42.8 ± 27.7 | 11.8 ± 18.0 | 47.0 ± 18.3 |
>
> Performance drops significantly in Settings B and C (ASR Both ≈ 10–35%), indicating that linear interpolation does not reliably preserve adversarial connectivity. In contrast, optimized Bézier paths achieve near-perfect connectivity, demonstrating that the improvement is not due to interpolation alone but requires the flexibility of the Bézier formulation. We will include this comparison in the final version.
>
> ### **W2) The intuition in Appendix A.1 is ReLU-specific and does not extend to ViT models.**
>
> We agree that the geometric intuition in Appendix A.1 is specific to ReLU networks and does not directly extend to architectures such as ViT. At the same time, our experimental results in Table 1 on ViT-B/16 show that adversarial connectivity and the effectiveness of Bézier crossover are also observed in this setting. This suggests that this phenomenon is not tied to ReLU-specific piecewise linearity, but reflects a broader structural property of modern deep networks. We will clarify this point in the paper and note that extending the theoretical understanding to transformer architectures is an important direction.
>
> ### **W3 & Q2) The reviewer points out an inconsistency with the monotonicity claim in Table 3.**
>
> While the overall trend shows that transferability generally improves with more auxiliary images, there are minor fluctuations across settings. We will revise the manuscript to replace “monotonically increases” with a more accurate description, such as “generally increases” or “shows an overall increasing trend.”
>
> ### **L1) Need to provide more concrete analysis or experiments on defense implications.**
>
>
> To examine the potential use of adversarial mode connectivity in a defense setting, we conduct a preliminary experiment within a standard adversarial training framework. We use a PGD-based adversarial training pipeline implemented with the MadryLab robustness library [1] on CIFAR-10 with a ResNet-18 model under the $\ell_\infty$ threat model ($\epsilon=8/255$). The baseline uses PGD with 5 steps and step size $\epsilon/4$. To incorporate connectivity, we modify the training loop by replacing standard PGD adversarial example generation in every 10th training iteration with samples generated along quadratic Bézier curves, while retaining standard PGD updates for the remaining iterations. For each input in the selected iterations, we generate two PGD endpoints, construct a quadratic Bézier curve between them, and select an interior point along the curve based on the loss. Both methods are trained for 30 epochs and evaluated under the same $\ell_\infty$ threat model with $\epsilon=8/255$ using 5-step PGD.
>
> | Method                 | Clean Acc (%) | Robust Acc (%) |
> |------------------------|---------------|----------------|
> | PGD                    | 68.39         | 42.33          |
> | Bézier-based (ours)    | 68.30         | **44.66**          |
>
> The Bézier-based variant improves robust accuracy while maintaining clean accuracy, suggesting that connectivity-based samples may provide complementary benefits in adversarial training. We thank the reviewer for this suggestion, which helped uncover an additional benefit of the Bézier crossover. We will include this experiment and discussion in the final version.
>
> [1] Engstrom et al., "Robustness (Python Library)," 2019. Available: https://github.com/MadryLab/robustness

---

> > ### Author Rebuttal · Reviewer_7mWj · 2026-04-01
> >
> > W1/Q1 and W3/Q2 are satisfactorily addressed. However, the geometric intuition in Appendix A.1 remains ReLU-specific and does not extend to ViT architectures used in the ImageNet experiments. I will maintain 5.
> > .

---

> > > ### Author Response · Authors · 2026-04-01
> > >
> > > We sincerely thank the reviewer for the positive feedback and for taking the time to read our rebuttal.
> > >
> > > Following the reviewer’s suggestion, we reconsidered the intuition from an architecture-agnostic perspective. Around a given image, modern classifiers often exhibit anisotropic loss geometry, where the loss changes sharply in some directions but more smoothly in others. Different attacks can therefore find distinct perturbations that lie within the same broad adversarial region. A straight line between them may leave that region, which helps explain why naive linear interpolation can fail, whereas an optimized Bézier path can bend to remain inside it. From this perspective, the key requirement is not a specific architecture, such as ReLU networks, but the presence of connected high-loss regions in the input loss landscape.

---

### Official Review · Reviewer_aRQ2 · 2026-03-10

**Soundness:** 2
**Presentation:** 3
**Significance:** 3
**Originality:** 3
**Overall Recommendation:** 4
**Confidence:** 3

**Summary:**

This study focuses on the evolutionary computation-based white-box adversarial attack against image classifiers. The proposed attack, MoCo-EA, leverages the concept of adversarial mode connectivity, which is a continuous path between distinct adversarial examples. The adversarial mode connectivity is defined as the Bezier curve, which is widely used in mode connectivity settings. MoCo-EA also utilizes the Bezier curve for its crossover operation to preserve the structure of the adversarial perturbation. Through experiments on CIFAR-10 and ImageNet, the existence of adversarial mode connectivity is tested based on the ratio of adversarial examples on the Bezier curve. MoCo-EA shows significant improvement compared to the traditional EA-based attacks. MoCo-EA shows competitive performance to existing gradient-based attacks for ResNet-50 trained on CIFAR-10 with defense mechanisms.

**Compliance With Llm Reviewing Policy:**

Affirmed.

**Final Justification:**

This study demonstrates the concept of adversarial mode connectivity and utilizes it to construct a stronger white-box attack. The adversarial mode connectivity helps to understand the relationship among different adversarial examples. This paper is well-organized and easy to follow. I had two major concerns about the computational cost and the scale of experiments before rebuttal. The authors’ rebuttal fully resolve these concerns. Thus, I raise my score to 4.

**Key Questions For Authors:**

- Does adversarial connectivity transfer among different victim models?
- How about the attack success rate of MoCo-EA against robust models on a larger test dataset?
- How many forward/backward operations are required to execute MoCo-EA? Are they larger than the existing gradient-based attacks?
- How about the attack transferability of MoCo-EA?

**Limitations:**

yes

**Strengths And Weaknesses:**

Strengths
- The concept of adversarial mode connectivity helps to understand the relationship among different adversarial examples. As shown in the experiments, this concept is useful to construct a stronger EA-based attack.
- This paper is well-organized and easy to follow.
- This study addresses how to leverage diverse adversarial examples to find another adversarial example efficiently. This is a promising way to develop sophisticated optimization algorithms for adversarial example generation.

Weaknesses
- Although the experiments cover a wide range of aspects for evaluation, the scale of each experiment is too small. For example, the experiment in Table 6 should be conducted with a larger number of test images because the attack performance depends on the test dataset. The remaining experiments should also be conducted with larger datasets.
- The computational cost of MoCo-EA seems to be too expensive compared to the gradient-based attacks. If the adversarial mode connectivity or adversarial example transfer across victim models, MoCo-EA is applicable to the black-box scenario, which is a strong advantage against other white-box attacks.

---

> ### Author Rebuttal · Authors · 2026-03-31
>
> We thank the reviewer for the helpful feedback and for noting that adversarial mode connectivity helps understand the relationship among adversarial examples. We address questions on the advantages of our method over gradient-based and black-box attacks below.
>
> ### **W1 & Q2) Experiments need to be conducted on larger test datasets, particularly for robust model evaluation in Table 6.**
>
> For the robust model setting on CIFAR-10, we evaluate on all 8,703 correctly classified images. For the obfuscated gradients setting on CIFAR-10, we evaluate on all 7,510 correctly classified images. The results are shown below:
>
> | Setting                 | PGD  | MIFGSM | AA   | AAA  | MoCo-EA |
> |-------------------------|------|--------|------|------|---------|
> | Robust model            | 39.6 | 38.7   | 40.3 | 39.1 | **40.5** |
> | Obfuscated gradients    | 13.5 | 13.3   | 0.0  | 12.8 | **53.8** |
>
> On the full evaluation set, MoCo-EA still achieves the highest attack success rate among the compared methods.
>
> ### **W2-1 & Q3) Computational cost and number of forward/backward operations compared to gradient-based attacks.**
> MoCo-EA is more expensive per generation than first-order gradient-based attacks. In our implementation, each generation with population size N requires $N + (\tau s + k)\lfloor N/2\rfloor$ forward passes and $\tau\lfloor N/2\rfloor$ backward calls, where $\tau$ is the number of control-point optimization steps, $s$ is the number of sampled $t$ values per step, and $k$ is the number of candidate points evaluated during offspring selection. Each backward call is computed over the mini-batch of $s$ sampled curve points, corresponding to $\tau s\lfloor N/2\rfloor$ single-point backward-equivalent computations.
> With our default setting ($\tau = 5$, $s = 3$, $k = 6$, $N = 30$), this amounts to 345 forward passes and 75 backward calls (equivalent to 225 single-point backward passes), compared to 40 forward and 40 backward passes for PGD-40.
> However, MoCo-EA’s higher per-generation cost reflects a deliberate trade-off: it allocates additional computation in geometry-aware exploration of the adversarial space, which yields benefits when gradient signals alone are insufficient, as reflected in Table 6: under obfuscated gradients, MoCo-EA achieves 32% ASR versus 16–17% for all gradient-based baselines.
> ### **W2-2 & Q1) Transferability of adversarial connectivity across victim models.**
>
> This experiment shows that adversarial connectivity transfers across models. On CIFAR-10, we optimize the Bézier path on a ResNet-18 source model and evaluate on a VGG-16 target model using the same metrics as Table 2.
>
> Norm | Endp. Avg | Path Succ. | Imgs Resc. | Avg Pts
> |--------------|--------------|--------------|--------------|--------------|
> $\ell_\infty$ | 10.8 ± 1.7 | 17.6 ± 2.8 | 3.1 ± 1.7 | 6.1 ± 1.2
> $\ell_2$ | 2.4 ± 1.0 | 4.0 ± 1.9 | 0.7 ± 0.9 | 1.1 ± 0.6
> $\ell_1$ | 2.0 ± 1.1 | 5.5 ± 2.1 | 2.4 ± 1.2 | 1.3 ± 0.5
>
> As shown above, connectivity-based path transferability consistently exceeds endpoint-average transferability across all three norms, and the nonzero $\textit{Imgs Resc.}$ values further indicate that intermediate points on the adversarial path can remain effective even when both endpoints fail on the target model. This finding suggests that adversarial connectivity can persist across architectures. We further support this with additional zeroth-order optimization experiments in a strict black-box setting. The zeroth-order Bézier variant achieves a 96.0% success rate on CIFAR-10.
>
> ### **Q4) How about the attack transferability of MoCo-EA?**
>
> We conduct a transferability experiment for MoCo-EA under the same protocol as Table 2. For each selected image or image pair, we run MoCo-EA, keep the endpoint perturbations it produces, record the crossover curves explored during evolution, sample perturbations from these curves, and evaluate their transferability to unseen images.
>
> | Setting | Norm | Endp. Avg  | Path Succ.  | Imgs Resc.  | Avg Pts    |
> | ------- | ---- | ---------- | ----------- | ----------- | ---------- |
> | A       | ℓ∞   | 25.6 ± 4.8 | 45.4 ± 8.9  | 14.1 ± 4.5  | 29.5 ± 6.7 |
> | A       | ℓ₂   | 5.9 ± 1.4  | 9.5 ± 2.2   | 2.6 ± 1.8   | 7.0 ± 1.7  |
> | A       | ℓ₁   | 11.7 ± 2.9 | 26.1 ± 21.6 | 16.4 ± 13.9 | 7.2 ± 5.9  |
> | B       | ℓ∞   | 24.9 ± 5.2 | 52.1 ± 7.7  | 17.4 ± 6.2  | 29.0 ± 6.4 |
> | B       | ℓ₂   | 5.9 ± 1.0  | 11.1 ± 1.4  | 2.8 ± 1.4   | 7.0 ± 1.0  |
> | B       | ℓ₁   | 11.0 ± 2.9 | 35.7 ± 16.5 | 22.7 ± 11.5 | 7.8 ± 4.4  |
> | C       | ℓ∞   | 37.6 ± 2.3 | 63.3 ± 4.0  | 17.3 ± 4.1  | 45.0 ± 2.0 |
> | C       | ℓ₂   | 7.4 ± 1.1  | 14.0 ± 3.1  | 3.6 ± 2.1   | 8.7 ± 1.0  |
> | C       | ℓ₁   | 11.9 ± 2.2 | 45.6 ± 5.8  | 27.5 ± 5.3  | 9.5 ± 3.6  |
>
> Overall, the results show that perturbations sampled from MoCo-EA’s explored curves transfer better than the endpoint perturbations alone across all norms and settings.

---

> > ### Author Rebuttal · Reviewer_aRQ2 · 2026-04-04
> >
> > Thank you for the authors' response to my questions and concerns. The authors provide empirical evidence to resolve my concerns listed as weaknesses and questions.

---

> > > ### Author Response · Authors · 2026-04-04
> > >
> > > Thank you for your thoughtful review. We are glad that our additional experiments addressed your concerns.
> > >
> > > Specifically, we conducted large-scale evaluations for both the robust and obfuscated gradient settings, provided a detailed analysis of computational costs, and demonstrated both the transferability across models and the transferability of MoCo-EA itself.
> > >
> > > We appreciate your valuable feedback.

---

### Official Review · Reviewer_R3U6 · 2026-03-13

**Soundness:** 3
**Presentation:** 3
**Significance:** 3
**Originality:** 3
**Overall Recommendation:** 4
**Confidence:** 3

**Summary:**

This paper proposes MoCo-EA, an evolutionary adversarial attack method that leverages adversarial mode connectivity. By replacing traditional discrete crossover with an optimized quadratic Bézier curve crossover, the authors claim to achieve near-perfect attack success rates on CIFAR-10 and ImageNet, along with a drastic reduction in both the number of queries and convergence generations. Furthermore, it continues to outperform certain gradient-based methods when evaluated against robust models and gradient obfuscation defenses.

**Compliance With Llm Reviewing Policy:**

Affirmed.

**Key Questions For Authors:**

（1）Given that MoCo-EA requires white-box access to compute gradients for the Bézier crossover, in what practical scenarios would an attacker prefer this method over highly optimized, purely gradient-based ensembles (e.g. AutoAttack), aside from the obfuscated gradient defense scenario?

（2）Have you considered or experimented with adapting the Bézier crossover to a strict black-box setting, perhaps by using zeroth-order optimization to estimate the gradients for the control point?

（3）The current method uses quadratic Bézier curves. Do you hypothesize that higher-order curves (with multiple control points) could discover even flatter adversarial basins and yield better transferability, or would the optimization overhead outweigh the benefits?

**Limitations:**

（1）Dependence on Gradients. The primary limitation is the reliance on gradient information for the crossover operator, which restricts the algorithm from being deployed in standard black-box environments.

（2）While evaluated on ResNet-18 and ViT-Base, it remains to be seen if the continuous adversarial manifolds behave similarly in highly quantized networks or non-differentiable models.

**Strengths And Weaknesses:**

Strengths:

（1）Applying the concept of mode connectivity to the input perturbation space is highly innovative and provides a fresh perspective on the geometric structure of adversarial examples.

（2）The proposed Bézier crossover dramatically improves the efficiency of evolutionary attacks. The reduction in the number of generations and queries compared to traditional EAs is impressive.

Weaknesses:

（1）Evolutionary algorithms are typically favored in black-box settings. Because MoCo-EA requires white-box (gradient information) access to optimize the Bézier curve's control point, it somewhat blurs the line between gradient-based and gradient-free attacks. The motivation for using an EA framework when full gradients are available could be stronger.

（2）Although the number of queries is reduced, optimizing the Bézier curve requires forward and backward passes during the crossover phase. The comparison of computational complexity could be more transparent regarding the true cost of these gradient steps versus simple forward passes in traditional EAs.

---

> ### Author Rebuttal · Authors · 2026-03-31
>
> We thank the reviewer for the feedback and for highlighting the use of mode connectivity. We address the questions below.
>
> ### **W1 & Q1) The motivation for using an EA framework with accessible gradients and its practical advantages over gradient-based attacks are unclear.**
>
> MoCo-EA performs multi-trajectory, population-based exploration, and operates over **continuous adversarial paths rather than isolated points in standard white-box methods**. This leads to three practical advantages beyond obfuscated-gradient scenarios. First, it improves transferability. As shown in Table 2, intermediate points along optimized curves consistently outperform endpoints, demonstrating that MoCo-EA discovers flatter, more universal adversarial directions. This advantage also persists in cross-model settings (see our response to Reviewer aRQ2, W2-2 & Q1). Second, MoCo-EA enables broader global exploration of the adversarial space. By maintaining a population and exploring connectivity between solutions, MoCo-EA can escape local optima that trap gradient-based methods. Third, MoCo-EA generates multiple solutions. Instead of producing a single perturbation, MoCo-EA generates an entire adversarial trajectory.
>
> ### **W2) Clearer comparison of computational cost between gradient steps and forward passes in traditional EAs.**
>
> In MoCo-EA, Bézier crossover introduces a small number of backward passes per generation, whereas traditional EAs rely primarily on forward evaluations. Concretely, each generation with population size $N$ requires $N + (\tau s + k)\lfloor N/2\rfloor$ forward passes and $\tau\lfloor N/2\rfloor$ backward calls (each over $s$ sampled curve points), where $\tau$ is the number of control-point optimization steps, $s$ the number of sampled $t$ values per step, and $k$ the number of candidate points evaluated during offspring selection. This is higher per generation than the baseline EA ($N$ forward passes).
> However, this overhead is offset by a substantial reduction in the number of generations and total queries needed to reach a successful attack (Table 5). MoCo-EA converges in ~1–2 generations versus hundreds for the baseline EA, resulting in lower total query cost and runtime despite the higher per-generation cost. We will add this breakdown to the revised appendix.
> ### **Q2 & L1) Adapting Bézier crossover to a strict black-box setting (e.g., via zeroth-order optimization).**
> We implement a black-box adaptation of the Bézier crossover on CIFAR-10 under the ℓ∞ norm (Table 5). We replace the gradient-based control-point update with a zeroth-order optimization.
>
> | Metric       | MoCo-EA (Bezier)  |
> | ------------ | ----------------- |
> | Succ. rate   | 96.0          |
> | Avg. gen.    | 242.9 ± 201.7 |
> | Avg. queries | 8296 ± 7451   |
> | Avg. time    | 24.44 ± 21.97 |
>
> This suggests that MoCo-EA can be applied in black-box settings.
>
> ### **Q3) Would higher-order Bézier curves improve transferability, or would the added optimization cost outweigh the benefits?**
>
> We conducted an ablation study replacing the quadratic Bézier path with a cubic Bézier path on CIFAR-10 in Table 2. In addition to the standard transferability metrics, we report $\textit{Opt.\ Time}$, which measures the additional curve-optimization cost per sample in seconds.
>
> | Setting | Norm | Endp. Avg  | Path Succ. | Imgs Resc. | Avg Pts    | Opt. Time |
> | ------- | ---- | ---------- | ---------- | ---------- | ---------- | --------- |
> | A       | ℓ∞   | 22.9 ± 3.8 | 40.8 ± 5.6 | 11.0 ± 3.7 | 15.1 ± 2.7 | 4.5 ± 0.1 |
> | A       | ℓ₂   | 6.4 ± 1.3  | 9.0 ± 2.4  | 1.0 ± 1.0  | 3.3 ± 0.7  | 4.6 ± 0.0 |
> | A       | ℓ₁   | 4.3 ± 1.2  | 11.4 ± 2.5 | 4.8 ± 1.8  | 3.7 ± 0.8  | 4.9 ± 0.0 |
> | B       | ℓ∞   | 24.2 ± 3.3 | 45.6 ± 5.7 | 13.6 ± 4.2 | 17.5 ± 2.5 | 8.8 ± 0.1 |
> | B       | ℓ₂   | 6.9 ± 1.1  | 10.8 ± 1.7 | 1.8 ± 1.3  | 3.7 ± 0.6  | 8.9 ± 0.1 |
> | B       | ℓ₁   | 4.6 ± 1.2  | 12.4 ± 1.6 | 5.6 ± 1.9  | 4.1 ± 0.8  | 9.1 ± 0.1 |
> | C       | ℓ∞   | 24.2 ± 2.2 | 42.2 ± 3.4 | 9.9 ± 3.1  | 15.6 ± 1.6 | 8.7 ± 0.1 |
> | C       | ℓ₂   | 7.6 ± 1.4  | 12.4 ± 2.9 | 2.0 ± 1.7  | 3.8 ± 1.0  | 8.9 ± 0.1 |
> | C       | ℓ₁   | 4.6 ± 1.3  | 13.0 ± 3.8 | 6.1 ± 3.5  | 4.2 ± 1.2  | 9.1 ± 0.1 |
>
> Compared to the quadratic variant, the cubic curve improves transferability in some cases (notably under ℓ∞), but the gains are not consistent across all settings and come with increased optimization cost.
>
> ### **L2) Behavior of adversarial connectivity in quantized or non-differentiable models.**
>
> We evaluate Setting A in Table 1 under quantized (low-bit) and non-differentiable (median filtering) variants.
>
> | Model       | ℓ∞    | ℓ₂    | ℓ₁    |
> | ----------- | ----- | ----- | ----- |
> | FP32        | 100.0 | 100.0 | 100.0 |
> | 4-bit act   | 100.0 | 100.0 | 100.0 |
> | 2-bit act   | 100.0 | 99.5  | 99.4  |
> | Median filt | 96.3  | 97.0  | 93.5  |
>
> These results suggest that adversarial path connectivity is still observed under quantized and non-differentiable transformations.

---

> > ### Author Rebuttal · Reviewer_R3U6 · 2026-04-02
> >
> > The authors’ response provides additional empirical results, which partially alleviate the concerns regarding black-box applicability and the design of higher-order curves. However, the rebuttal remains insufficient in clarifying the method’s distinct practical value over mature purely gradient-based attacks when full white-box gradients are available, as well as in establishing a fair comparison between query reduction and the true computational cost.

---

> > > ### Author Response · Authors · 2026-04-03
> > >
> > > We sincerely thank the reviewer for the feedback and for taking the time to read our rebuttal.
> > >
> > > ### **Q1) Practical value over gradient-based attacks.**
> > >
> > > We clarify that the key advantage of MoCo-EA is not improved point-wise optimization over gradient-based attacks, but structured global exploration beyond single-trajectory optimization.
> > >
> > > Specifically, MoCo-EA is preferable in the following practical scenarios:
> > >
> > > (1) When multiple diverse adversarial solutions are required.
> > >
> > > Gradient-based attacks typically produce a single perturbation tied to one optimization trajectory. In contrast, MoCo-EA maintains a population and generates an entire set of distinct successful perturbations along trajectories. This is useful in scenarios such as robustness evaluation and adversarial training, where relying on a single solution can underestimate vulnerability.
> > >
> > > The preliminary experiment shows that incorporating connectivity-based samples can improve robustness while maintaining clean accuracy, suggesting that these perturbations capture complementary adversarial directions beyond those discovered by standard gradient-based attacks. (For details, please see our response to Reviewer 7mWj, L1).
> > >
> > > | Method                 | Clean Acc (%) | Robust Acc (%) |
> > > |------------------------|---------------|----------------|
> > > | PGD                    | 68.39         | 42.33          |
> > > | Bézier-based (ours)    | 68.30         | **44.66**          |
> > >
> > > (2) When transferability is critical (e.g., cross-model or downstream attacks).
> > >
> > > We consistently observe that path-based perturbations outperform endpoint-based perturbations. In Table 2, intermediate points along optimized Bézier paths achieve higher success rates than the PGD endpoints.
> > >
> > > We further extend this to a cross-model setting (ResNet-18 → VGG-16), where path success consistently exceeds endpoint success across all norms, indicating that these perturbations generalize beyond the source model.
> > >
> > > Norm | Endp. Avg | Path Succ. | Imgs Resc. | Avg Pts
> > > |--------------|--------------|--------------|--------------|--------------|
> > > $\ell_\infty$ | 10.8 ± 1.7 | 17.6 ± 2.8 | 3.1 ± 1.7 | 6.1 ± 1.2
> > > $\ell_2$ | 2.4 ± 1.0 | 4.0 ± 1.9 | 0.7 ± 0.9 | 1.1 ± 0.6
> > > $\ell_1$ | 2.0 ± 1.1 | 5.5 ± 2.1 | 2.4 ± 1.2 | 1.3 ± 0.5
> > >
> > > (3) When escaping local optima matters despite full gradient access.
> > >
> > > Even in white-box settings, gradient-based attacks follow a single trajectory and can converge to local optima that do not generalize well. MoCo-EA instead performs structured population-based exploration over connected regions in the loss landscape. As shown in Table 6, this leads to higher success rates on robust models, indicating that the benefit comes from global exploration.
> > >
> > > (4) When we need to attack under obfuscated gradients.
> > >
> > > We additionally ran a large-scale evaluation on CIFAR-10 of 7,510 correctly classified images. MoCo-EA significantly outperforms all baselines. The results are shown below:
> > >
> > > | Setting                 | PGD  | MIFGSM | AA   | AAA  | MoCo-EA |
> > > |-------------------------|------|--------|------|------|---------|
> > > | Obfuscated gradients    | 13.5 | 13.3   | 0.0  | 12.8 | **53.8** |
> > >
> > > Overall, MoCo-EA is not intended to replace gradient-based attacks, but is preferable when the objective involves diversity, transferability, or structural exploration beyond single-point optimization.
> > >
> > > ### **Q2) Establish a fair comparison between query reduction and the true computational cost.**
> > >
> > > We want to clarify that Table 5 reflects the overall computational efficiency of MoCo-EA. In particular, “Avg. queries” counts all model forward evaluations (including those used during Bézier optimization), and “Avg. time” reports the end-to-end runtime. These results show that MoCo-EA significantly reduces both the number of evaluations and the total runtime compared to traditional EA.
> > >
> > > To further clarify the cost, we additionally provide a breakdown of the computation beyond what is reported in Table 5. In particular, we analyze 1) external queries (population and candidate evaluations only) and 2) backward calls introduced by Bézier control-point optimization.
> > >
> > > | Metric                        | Traditional EA | MoCo-EA (Bézier) |
> > > | ----------------------------- | -------------- | ---------------- |
> > > | External queries              | 12,329         | **234**          |
> > > | Backward calls                | 0              | 127              |
> > >
> > > As shown in the table, MoCo-EA significantly reduces the number of external queries (12,329 → 234). Despite the additional backward calls (127 on average), the overall runtime remains lower, as reflected in Table 5 (“Avg. time”).

---

### Official Review · Reviewer_GoKY · 2026-03-19

**Soundness:** 3
**Presentation:** 2
**Significance:** 3
**Originality:** 3
**Overall Recommendation:** 4
**Confidence:** 4

**Summary:**

The paper proposes MoCo-EA, a method that combines evolutionary algorithms (EA) with the concept of mode connectivity to find adversarial perturbations. Specifically, it replaces traditional discrete crossover operators with a Bezier-path-based crossover, arguing that successful perturbations lie on connected low-loss manifolds.

While the paper demonstrates empirical gains over traditional EA baselines, the core contribution appears to be incremental "A+B" combination (Evolutionary Algorithms + Mode Connectivity) without sufficient justification for its unique advantages or a fair comparison against established white-box attacks.

**Compliance With Llm Reviewing Policy:**

Affirmed.

**Final Justification:**

All my concerns have been addressed. It is observable that the performance gain of MoCo-EA over Traditional EA is significantly greater than its advantage over AAA in scenarios with normal gradients. These points need to be discussed more clearly in the final draft. At this stage, I am willing to recommend a 'Weak Accept'.

**Key Questions For Authors:**

1. Since this is a white-box attack with accessible gradients, what is the specific advantage of using an evolutionary framework over direct manifold optimization or advanced gradient-based transfer methods?

2. Can you provide a comparison against state-of-the-art gradient-based attacks under a matched computational budget (time and FLOPs) rather than just comparing against a weak EA baseline?

**Limitations:**

1. The innovation is largely incremental rather than a fundamental shift in new adversarial theory.
2. Applying EA to a white-box setting is unconventional and lacks a strong motivation compared to standard optimization.

**Strengths And Weaknesses:**

### Pros
1. Reframes the crossover operator as a geometry-aware path exploration rather than random recombination, which is an interesting conceptual shift.
2. Demonstrates significant efficiency gains (lower query counts and faster runtime) compared specifically to the chosen traditional evolutionary baseline.

### Cons

1. The methodology feels like a straightforward "stitching" of two existing concepts: evolutionary algorithms and mode connectivity. There is a lack of fundamental insight into why this specific combination is necessary or superior to more direct optimization techniques.
2. Evolutionary algorithms are typically leveraged for black-box scenarios where gradients are unavailable. However, this paper applies EA to a white-box setting. Given that gradients are accessible, the paper fails to justify why an EA-based approach is preferable to standard, more efficient gradient-based optimization with similar computational complexity.
3. The method relies on gradient-based endpoint generation and gradient-based optimization of the Bezier control points. This makes it a hybrid gradient-assisted method rather than a pure evolutionary breakthrough, further diluting the significance of the EA framework.

---

> ### Author Rebuttal · Authors · 2026-03-31
>
> We thank the reviewer for the feedback and for finding our method interesting. We also appreciate the question regarding the introduction of mode connectivity into an evolutionary framework in a white-box setting.
>
> ### **W1 & L1) The method is viewed as a straightforward combination of evolutionary algorithms and mode connectivity.**
>
> We respectfully disagree that our method is a simple "A+B" combination of evolutionary algorithms and mode connectivity. The key contribution is a **new optimization paradigm with geometry-aware population search**, rather than a modular integration of two existing ideas.
>
> Specifically, prior evolutionary attacks treat perturbations as independent points and rely on **discrete, structure-agnostic crossover**, which often destroys adversarial properties. In contrast, our work introduces a continuous, optimization-driven crossover operator that operates along adversarial manifolds. This significantly changes the role of crossover from random mixing to **structured exploration of connected high-loss regions**. This shift is not achievable by either component alone: (1) Mode connectivity alone provides analysis, but not a scalable search mechanism; (2) Evolutionary algorithms alone lack geometry awareness and thus suffer from inefficient exploration. Our results empirically validate that this combination yields consistent improvements: intermediate points along optimized paths consistently outperform endpoints in transferability, revealing a **previously unexploited structure of adversarial space**.
>
> ### **W2 & Q1 & L2) The motivation for using an EA framework in a white-box setting with accessible gradients is unclear.**
>
> The key advantage of the evolutionary framework in our setting is **not the absence of gradients, but the ability to perform structured global exploration beyond single-trajectory optimization**.
>
> Gradient-based attacks follow a **single optimization trajectory**, which limits exploration and often converges to sharp, localized adversarial maxima with limited transferability. In contrast, MoCo-EA maintains a **population of diverse perturbations** and explores connected regions of the loss landscape, enabling (1) robustness to gradient issues, including obfuscation or instability; (2) escape from local maxima through population diversity; and (3) discovery of flatter, more transferable adversarial regions via path-based optimization. This distinction is reflected in our empirical results. For example, under gradient-obfuscated settings, where gradient-based methods degrade significantly, MoCo-EA remains effective and consistently outperforms them (Table 6). This demonstrates that the evolutionary framework provides **complementary strengths**, even in white-box settings. More broadly, our goal is not to replace gradient-based attacks, but to introduce a new search mechanism that leverages both global exploration and local geometry.
>
> ### **W3) The method is viewed as a hybrid gradient-assisted approach, which weakens the role of the EA framework.**
> We respectfully clarify that incorporating gradients does not diminish the evolutionary nature of our method, but instead enhances it by enabling geometry-aware recombination. MoCo-EA retains all core components of evolutionary algorithms, and its search dynamics fundamentally rely on population-based exploration rather than gradient descent. Gradients are used only locally to optimize the control point of the crossover path, not to directly optimize adversarial perturbations end-to-end. This design is intentional: traditional crossover is the main bottleneck in evolutionary attacks, as it disrupts adversarial structure. By introducing gradient-informed path optimization, we transform crossover into a constructive operator that preserves and enhances adversarial properties. In this sense, gradients are not replacing EA, but making its key operator (crossover) effective for the first time in this context.
> ### **Q2) Comparison under matched computational budgets.**
>
> We provide a matched-compute comparison with gradient-based baselines on 100 randomly selected clean-correct CIFAR-10 test images. We match average FLOPs across methods on the same image subset.
>
> | Setting              | PGD   | MIFGSM | AA   | AAA  | MoCo-EA |
> |----------------------|-------|--------|------|------|---------|
> | Obfuscated gradients | 14.0% | 17.0%  | 0.0% | 9.0% | **64.0%** |
>
> Even under this setting, MoCo-EA outperforms gradient-based attacks. These results suggest that factors beyond the optimization budget contribute to the advantage of MoCo-EA. While MoCo-EA includes gradient-based updates during Bézier crossover, it is designed to improve search efficiency rather than minimize FLOPs.

---

> > ### Author Rebuttal · Reviewer_GoKY · 2026-04-03
> >
> > Thanks for your clarification which partially resolved my concern on the novelty. I have follow-up questions for the comparison to gradient-based attacks:
> >
> > 1. Could author provide attack results for adversarially trained (or any other robust) models on ImageNet[1,2,3]? The current difference on cifar-10 is marginal.
> > 2. For obfuscated gradients, could the author provide other baselines beyond PGD-like gradient-based attack, such as black-box based evolutionary algorithms and corresponding adaptive attacks.
> >
> > I believe that adding more discussion with stronger baselines will improve the overall quality of this draft. If the author can address my concerns above, I can consider increasing my score further. For now, I'm willing to give a weak reject.
> >
> > [1] Feature Denoising for Improving Adversarial Robustness
> >
> > [2] https://github.com/Hadisalman/smoothing-adversarial
> >
> > [3] https://robustbench.github.io/#div_imagenet_Linf_heading

---

> > > ### Author Response · Authors · 2026-04-04
> > >
> > > We sincerely thank the reviewer for the feedback and for taking the time to read our rebuttal carefully.
> > >
> > > ### **Q1) Provide attack results for adversarially trained (or any other robust) models on ImageNet.**
> > >
> > > Following the reviewer’s suggestion, we conducted additional experiments on adversarially trained ImageNet models using a standard ℓ∞ setting.
> > >
> > > We use the adversarially trained ResNet-50 model from RobustBench (Salman et al., 2020, Salman2020Do_R50), which is a benchmark for ImageNet robustness. All methods are evaluated on the same 100 randomly sampled test images that are correctly classified.
> > >
> > >
> > > | Setting                 | PGD  | MIFGSM | AA   | AAA  | MoCo-EA |
> > > |-------------------------|------|--------|------|------|---------|
> > > | Robust model            | 38 | 38   | 38 | 43 | **47** |
> > >
> > > MoCo-EA outperforms all baselines, achieving a +4% improvement over Adaptive AutoAttack and +9% over standard gradient-based attacks.
> > >
> > > These results support our claim that MoCo-EA provides benefits beyond standard gradient-based optimization, particularly in exploring diverse adversarial regions that are harder to capture with single-trajectory methods.
> > >
> > > ### **Q2) For obfuscated gradients, provide other baselines beyond PGD-like gradient-based attack, such as black-box based evolutionary algorithms and corresponding adaptive attacks.**
> > >
> > > Following the reviewer’s suggestion, we extend the comparison to include both black-box evolutionary baselines and adaptive gradient-free attacks under the obfuscated-gradient setting. In particular, we additionally include: Traditional EA, Square Attack (score-based randomized search), and Natural Evolution Strategies (NES) (gradient-free estimation via stochastic sampling).
> > >
> > > We evaluate on 100 randomly sampled clean correctly classified CIFAR-10 images under the ℓ∞ threat model (ε = 8/255).
> > >
> > > | Setting     | PGD  | MIFGSM | AA  | AAA  | Square | NES  | Traditional EA | MoCo-EA |
> > > |-------------|------|--------|-----|------|--------|------|----------------|---------|
> > > | Obfuscated  | 17%  | 12%    | 0%  | 11%  | 36%    | 30%  | 42%            | **54%** |
> > >
> > > While gradient-based methods (PGD, MIFGSM, AA, AAA) largely fail, and black-box methods (Square, NES, Traditional EA) partially recover performance, MoCo-EA consistently achieves the highest success rate. This indicates that structured, connectivity-driven exploration provides a stronger mechanism for navigating obfuscated loss landscapes than both gradient-based and conventional gradient-free approaches.

---

### Decision · Program_Chairs · 2026-04-30

**Decision:**

Accept (regular)

**Comment:**

The AC has read all the reviews and the authors' responses. The rebuttal helps address many of the reviewers' concerns, including black-box applicability, computational cost compared to gradient-based methods, and evaluation on larger datasets. After the rebuttal, all reviewers are inclined to accept the work. The recommendation is acceptance. The authors should carefully revise the paper based on the reviewers' comments.